# Unsupervised Model Selection for Time-series Anomaly Detection

**Mononito Goswami & Cristian Challu**[*]
Carnegie Mellon University
{mgoswami,cchallu}@cs.cmu.edu

**Laurent Callot, Lenon Minorics & Andrey Kan**
Amazon Research
{lcallot,avkan}@amazon.com,
minorics@amazon.de

## Abstract

Anomaly detection in time-series has a wide range of practical applications. While numerous anomaly detection methods have been proposed in the literature, a recent survey concluded that no single method is the most accurate across various datasets. To make matters worse, anomaly labels are scarce and rarely available in practice. The practical problem of selecting the most accurate model for a given dataset without labels has received little attention in the literature. This paper answers this question *i.e.* Given an unlabeled dataset and a set of candidate anomaly detectors, how can we select the most accurate model? To this end, we identify three classes of surrogate (unsupervised) metrics, namely, *prediction error*, *model centrality*, and *performance on injected synthetic anomalies*, and show that some metrics are highly correlated with standard supervised anomaly detection performance metrics such as the $F_1$ score, but to varying degrees. We formulate metric combination with multiple imperfect surrogate metrics as a robust rank aggregation problem. We then provide theoretical justification behind the proposed approach. Large-scale experiments on multiple real-world datasets demonstrate that our proposed unsupervised approach is as effective as selecting the most accurate model based on partially labeled data.

## 1 Introduction

Anomaly detection in time-series data has gained considerable attention from the academic and industrial research communities due to the explosion in the amount of data produced and the number of automated system requiring some form of monitoring. A large number of anomaly detection methods have been developed to solve this task (Schmidl et al., 2022; Blázquez-García et al., 2021), ranging from simple algorithms (Keogh et al., 2005; Ramaswamy et al., 2000) to complex deep-learning models (Xu et al., 2018; Challu et al., 2022). These models have significant variance in performance across datasets (Schmidl et al., 2022; Paparrizos et al., 2022b), and evaluating their actual performance on real-world anomaly detection tasks is non-trivial, even when labeled datasets are available (Wu & Keogh, 2021).

Labels are seldom available for many, if not most, anomaly detection tasks. Labels are indications of which time points in a time-series are anomalous. The definition of an anomaly varies with the use case, but these definitions have in common that anomalies are rare events. Hence, accumulating a sizable number of labeled anomalies typically requires reviewing a large portion of a dataset by a domain expert. This is an expensive, time-consuming, subjective, and thereby error-prone task which is a considerable hurdle for labeling even a subset of data. Unsurprisingly, a large number of time-series anomaly detection methods are unsupervised or semi-supervised – *i.e.* they do not require any anomaly labels during training and inference.

There is no single universally best method (Schmidl et al., 2022; Paparrizos et al., 2022b). Therefore it is important to select the most accurate method for a given dataset without access to anomaly labels. The problem of unsupervised anomaly detection model selection has been overlooked in the literature, even though it is a key problem in practical applications. Thus, we offer an answer to the

---

[*]Work carried out when the first two authors were interns at Amazon AWS AI Labs.

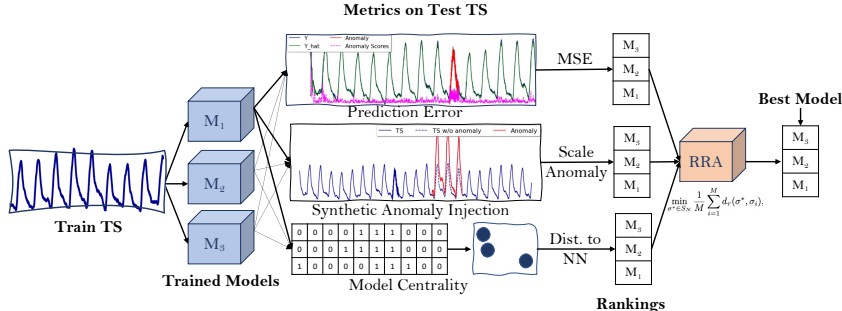

Figure 1: *The Model Selection Workflow*. We identify three classes of surrogate metrics of model quality (Sec. 3), and propose a novel robust rank aggregation framework to combine multiple rankings from metrics (Sec. 4).

question: **Given an unlabeled dataset and a set of candidate models, how can we select the most accurate model?**

Our approach is based on computing "surrogate" metrics that correlate with model performance yet do not require anomaly labels, followed by aggregating model ranks induced by these metrics (Fig. 1). Empirical evaluation on 10 real-world datasets spanning diverse domains such as medicine, sports, etc. shows that **our approach can perform unsupervised model selection as efficiently as selection based on labeling a subset of data**.

In summary, our contributions are as follows:

- To the best of our knowledge, we propose one of the first methods for unsupervised selection of anomaly detection models on time-series data. To this end, we identify intuitive and effective unsupervised metrics for model performance. Prior work has used a few of these unsupervised metrics for problems other than time-series anomaly detection model selection.

- We propose a novel robust rank aggregation method for combining multiple surrogate metrics into a single model selection criterion. We show that our approach performs on par with selection based on labeling a subset of data.

- We conduct large-scale experiments on over 275 diverse time-series, spanning a gamut of domains such as medicine, entomology, etc. with 5 popular and widely-used anomaly detection models, each with 1 to 4 hyper-parameter combinations, resulting in over 5,000 trained models. Upon acceptance, we will make our code publicly available.

In the next section, we formalize the model selection problem. We then describe the surrogate metric classes in Sec. 3, and present our rank aggregation method in Sec. 4. Our empirical evaluation is described in Sec. 5. Finally, we summarize related work in Sec. 6 and conclude in Sec. 7.

## 2 PRELIMINARIES & THE MODEL SELECTION PROBLEM

Let $\{\boldsymbol{x}_t, y_t\}_{t=1}^T$ denote a multivariate time-series (TS) with observations $(\boldsymbol{x}_1, \ldots, \boldsymbol{x}_T)$, $\boldsymbol{x}_t \in \mathbb{R}^d$ and anomaly labels $(y_1, \ldots, y_T)$, $y_t \in \{0, 1\}$, where $y_t = 1$ indicates that the observation $\boldsymbol{x}_t$ is an anomaly. The labels are only used for evaluating our selection approaches, but not for the model selection procedure itself.

Next, let $\mathcal{M} = \{A_i\}_{i=1}^N$ denote a set of $N$ candidate anomaly detection models. Each model $A_i$ is a tuple (`detector`, `hyper-parameters`), i.e., a combination of an anomaly detection method (e.g. LSTM-VAE (Park et al., 2018)) and a fully specified hyper-parameter configuration (e.g. for LSTM-VAE, `hidden_layer_size=128, num_layers=2, ⋯`).

Here, we only consider models that do not require anomaly labels for training. Some models still require unlabeled time-series as training data. We therefore, consider a train/test split $\{\boldsymbol{x}_t\}_{t=1}^{t_{\text{test}}-1}$,

$\{\boldsymbol{x}_t\}_{t=t_{\text{test}}}^T$, where the former is used if a model requires training (without labels). In our notation, $A_i$ denotes a trained model.

We assume that a trained model $A_i$, when applied to observations $\{\boldsymbol{x}_t\}_{t=t_{\text{test}}}^T$, produces anomaly scores $\{s_t^i\}_{t=t_{\text{test}}}^T$, $s_t^i \in \mathbb{R}_{\geq 0}$. We assume that a higher anomaly score indicates that the observation is more likely to be an anomaly. However, we do not assume that the scores correspond to likelihoods of any particular statistical model or that the scores are comparable across models.

Model performance or, equivalently, quality of the scores can be measured using a supervised metric $\mathcal{Q}(\{s_t\}_{t=1}^T, \{y_t\}_{t=1}^T)$, such as the Area under Precision Recall curve or best $F_1$ score, commonly used in literature. We discuss the choice of the quality metric in the next section.

In general, rather than considering a single time-series, we will be performing model selection for a set of $L$ time-series with observations $\mathcal{X} = \{\{\boldsymbol{x}_t^j\}_{t=1}^T\}_{j=1}^L$ and labels $\mathcal{Y} = \{\{y_t^j\}_{t=1}^T\}_{j=1}^L$, where $j$ indexes time-series. Let $\mathcal{X}_{\text{train}}$, $\mathcal{X}_{\text{test}}$, and $\mathcal{Y}_{\text{test}}$ denote the train and test portions of observations, and the test portion of labels, respectively. We are now ready to introduce the following problem.

**Problem 1.** *Unsupervised Time-series Anomaly Detection Model Selection. Given observations* $\mathcal{X}_{test}$ *and a set of models* $\mathcal{M} = \{A_i\}_{i=1}^N$ *trained using* $\mathcal{X}_{train}$, *select a model that maximizes the anomaly detection quality metric* $\mathcal{Q}(A_i(\mathcal{X}_{test}), \mathcal{Y}_{test})$. *The selection procedure cannot use labels.*

## 2.1 MEASURING ANOMALY DETECTION MODEL PERFORMANCE

Anomaly Detection can be viewed as a binary classification problem where each time point is classified as an anomaly or a normal observation. Hence, the performance of a model $A_i$ can be measured using standard precision and recall metrics. However, these metrics ignore the sequential nature of time-series; thus, time-series anomaly detection is usually evaluated using adjusted versions of precision and recall (Paparrizos et al., 2022a). We adopt widely used adjusted versions of precision and recall (Xu et al., 2018; Challu et al., 2022; Shen et al., 2020; Su et al., 2019; Carmona et al., 2021). These metrics treat time points as independent samples, except when an anomaly lasts for several consecutive time points. In this case, detecting any of these points is treated as if all points inside the anomalous segment were detected.

Adjusted precision and recall can be summarized in a metric called adjusted $F_1$ score, which is the harmonic mean of the two. The adjusted $F_1$ score depends on the choice of a decision threshold on anomaly scores. In line with several recent studies, we consider threshold selection as a problem orthogonal to our problem (Schmidl et al., 2022; Laptev et al., 2015; Blázquez-García et al., 2021; Rebjock et al., 2021; Paparrizos et al., 2022a;b) and therefore, consider metrics that summarize model performance across all possible thresholds. Common metrics include area under the precision-recall curve and best $F_1$ (maximum $F_1$ over all possible thresholds). Like Xu et al. (2018), we found a strong correlation between the two (App. A.11). We also found best $F_1$ to have statistically significant positive correlation with volume under the surface of ROC curve, a recently proposed robust evaluation metric (App. A.10). Thus, in the remainder of the paper, we restrict our analysis to identifying models with the highest *adjusted best $F_1$* score (*i.e.* $\mathcal{Q}$ is *adjusted best $F_1$*).

## 3 SURROGATE METRICS OF MODEL PERFORMANCE

The problem of unsupervised model selection is often viewed as finding an unsupervised metric which *correlates* well with supervised metrics of interest (Ma et al., 2021; Goix, 2016; Lin et al., 2020; Duan et al., 2019). Each unsupervised metric serves as a noisy measure of "model goodness" and reduces the problem of picking the best performing model according to the metric. We identified three classes of *imperfect* metrics that closely align with expert intuition in predicting the performance of time-series anomaly detection models. Our metrics are unsupervised because *they do not require anomaly labels*. However, some metrics such as $F_1$ score on synthetic anomalies are typically used for supervised evaluation. To avoid confusion we use term *surrogate* for our metrics. Below we elaborate on each class of surrogate metrics, starting with an intuition behind each class.

**Prediction Error** *If a model can forecast or reconstruct time-series well it must also be a good anomaly detector.* A large number of anomaly detection methods are based on forecasting or reconstruction (Schmidl et al., 2022), and for such models, we can compute forecasting or recon-

struction error without anomaly labels. We collectively call these *prediction error* metrics and consider a common set of statistics, such as mean absolute error, mean squared error, mean absolute percentage error, symmetric mean absolute percentage error, and the likelihood of observations. For multivariate time-series, we average each metric across all the variables. For example, given a series of observations $\{x_t\}_{t=1}^{T}$ and their predictions $\{\hat{x}_t\}_{t=1}^{T}$, mean squared error is defined as $\text{MSE} = \frac{1}{T}\sum_{t=1}^{T}(x_t - \hat{x}_t)^2$. In the interest of space, we refer the reader to a textbook (e.g., Hyndman & Athanasopoulos (2018)) for definitions of other metrics. Prior work on time-series anomaly detection (Saganowski & Andrysiak, 2020; Laptev et al., 2015; Kuang et al., 2022) used forecasting error to select between some classes of models. Prediction metrics are not perfect for model selection because the time-series can contain anomalies. Low prediction errors are desirable for all points except anomalies, but we do not know where anomalies are without labels. While prediction metrics can only be computed for anomaly detection methods based on time-series forecasting or reconstruction, the next two classes of metrics apply to any anomaly detection method.

**Synthetic Anomaly Injection** *A good anomaly detection model will perform well on data with synthetically injected anomalies.* Some studies have previously explored using synthetic anomaly injection to train anomaly detection models (Carmona et al., 2021). Here, we extend this line of research by systematically evaluating model selection capability after injecting different kinds of anomalies. Given an input time-series without anomaly labels, we randomly inject an anomaly of a particular type. The location of the injected anomaly is then treated as a (pseudo-)positive label, while the rest of the time points are treated as a (pseudo-)negative label. We then select a model that achieves the *best adjusted $F_1$* score with respect to pseudo-labels. Instead of relying on complex deep generative models (Wen et al., 2020) we devise simple and efficient yet effective procedures to inject anomalies of previously

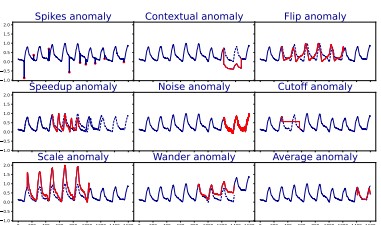

Figure 2: *Different kinds of synthetic anomalies.* We inject 9 different types of synthetic anomalies randomly, one at a time, across multiple copies of the original time-series. The - - is original time-series before anomalies are injected, – is the injected anomaly and the solid – is the time-series after anomaly injection.

described types (Schmidl et al., 2022; Wu & Keogh, 2021) as shown in Fig. 2. We defer the details of anomaly injection approaches to Appendix A.5. Model selection based on anomaly injection might not be perfect because (i) real anomalies might be of a different type compared to injected ones, and (ii) there may already be anomalies for which our pseudo-labels will be negative.

**Model Centrality** *There is only one ground truth, thus models close to ground truth are close to each other. Hence, the most "central" model is the best one.* Centrality-based metrics have seen recent success in unsupervised model selection for disentangled representation learning (Lin et al., 2020; Duan et al., 2019) and anomaly detection (Ma et al., 2021). To adapt this approach to time-series, we leverage the scores from the anomaly detection models to define a notion of model proximity. Anomaly scores of different models are not directly comparable. However, since anomaly detection is performed by thresholding the scores, we can consider ranking of time points by their anomaly scores. Let $\sigma_k(i)$ denote the rank of time point $i$ according to the scores from model $k$. We define the distance between models $A_k$ and $A_l$ as the Kendall's $\tau$ distance, which is the number of pairwise disagreements:

$$d_\tau(\sigma_k, \sigma_l) = \sum_{i<j} \mathbb{I}\{(\sigma_k(i) - \sigma_k(j))(\sigma_l(i) - \sigma_l(j)) < 0\} \tag{1}$$

Next, we measure model centrality as the average distance to its $K$ nearest neighbors, where $K$ is a parameter. This metric favors models which are close to their nearest neighbors. The centrality-based approach is imperfect, because "bad" models might produce similar results and form a cluster.

Each metric, while imperfect, is predictive of anomaly detection performance but to a varying extent as we show in Sec. 5. Access to multiple noisy metrics raises a natural question: *How can we combine multiple imperfect metrics to improve model selection performance?*

## 4 ROBUST RANK AGGREGATION

Many studies have explored the benefits of combining multiple sources of noisy information to reduce error in different areas of machine learning such as crowd-sourcing (Dawid & Skene, 1979) and programmatic weak supervision (Ratner et al., 2016; Goswami et al., 2021; Dey et al., 2022; Gao et al., 2022; Zhang et al., 2022a). Each of our surrogate metrics induces a (noisy) model ranking. Thus two natural questions arise in the context of our work: (1) *How do we reliably combine multiple model rankings?* (2) *Does aggregating multiple rankings help in model selection?*

### 4.1 APPROACHING MODEL SELECTION AS A RANK AGGREGATION PROBLEM

Let $[N] = \{1, 2, \cdots, N\}$ denote the universe of elements and $S_N$ be the set of permutations on $[N]$. For $\sigma \in S_N$, let $\sigma(i)$ denote the rank of element $i$ and $\sigma^{-1}(j)$ denote the element at rank $j$. Then the rank aggregation problem is as follows.

**Problem 2. *Rank Aggregation.*** *Given a collection of $M$ permutations $\sigma_1, \cdots, \sigma_M \in S_N$ of $N$ items, find $\sigma^* \in S_N$ that best summarizes these permutations according to some objective $C(\sigma^*, \sigma_1, \cdots, \sigma_M)$.*

In our context, the $N$ items correspond to the $N$ candidate anomaly detection models, $M$ corresponds to the number of surrogate metrics, $\sigma^*$ is the best summary ranking of models, and $\sigma^{*-1}(1)$ is the predicted best model.[1]

The problem of rank aggregation has been extensively studied in social choice, bio-informatics and machine learning literature. Here, we consider *Kemeny rank aggregation* which involves choosing a distance on the set of rankings and finding a barycentric or median ranking (Korba et al., 2017). Specifically, the rank aggregation problem is known as the Kemeny-Young problem if the aggregation objective is defined as $C = \frac{1}{M} \sum_{i=1}^{M} d_\tau(\sigma^*, \sigma_i)$, where $d_\tau$ is the Kendall's $\tau$ distance (Eq. 1).

Kemeny aggregation has been shown to satisfy many desirable properties, but it is NP-hard even with as few as 4 rankings (Dwork et al., 2001). To this end, we consider an efficient approximate solution using the *Borda method* (Borda, 1784) in which each item (model) receives points from each ranking (surrogate metric). For example, if a model is ranked $r$-th by a surrogate metric, it receives $(N - r)$ points from that metric. The models are then ranked in descending order of total points received from all metrics.

Suppose that we have several surrogate metrics for ranking anomaly detection models. Why would it be beneficial to aggregate the rankings induced by these metrics? The following theorem provides an insight into the benefit of Borda rank aggregation.

Consider two arbitrary models $i$ and $j$ with (unknown) ground truth rankings $\sigma^*(i)$ and $\sigma^*(j)$. Without the loss of generality, assume $\sigma^*(i) < \sigma^*(j)$, i.e., model $i$ is better. Next, let's view each surrogate metric $k = 1, \ldots, M$ as a random variable $\Omega_k$ taking values in permutations $S_N$, such that $\Omega_k(i) \in [N]$ denotes the rank of item $i$. We apply Borda rank aggregation on realizations of these random variables. Theorem 1 states that, under certain assumptions, the probability of Borda ranking making a mistake (i.e., placing model $i$ lower than $j$) is upper bounded in a way such that increasing $M$ decreases this bound. In appendix A.14, we empirically show that model selection performance improves as we increase the number of surrogate metrics (rankings).

**Theorem 1.** *Assume that $\Omega_k$'s are pairwise independent, and that $\mathbb{P}(\Omega_k(i) < \Omega_k(j)) > 0.5$ for any arbitrary models $i$ and $j$, i.e., each surrogate metric is better than random. Then the probability that Borda aggregation makes a mistake in ranking items $i$ and $j$ is at most $2 \exp\left(-\frac{M\epsilon^2}{2}\right)$, for some fixed $\epsilon$.*

The proof of the theorem is given in Appendix A.8. We do not assume that $\Omega_k$ are identically distributed, but only that $\Omega_k$'s are pairwise independent. We emphasize that the notion of independence of $\Omega_k$ as random variables is different from rank correlation between realizations of these variables (i.e., permutations). For example, two permutations drawn independently from a Mallow's distribution can have a high Kendall's $\tau$ coefficient (rank correlation). Therefore, our assumption of

---

[1]We use terms "ranking" and "permutation" as synonyms. The subtle difference is that in a ranking multiple models can receive the same rank. Where necessary, we break ties uniformly at random.

independence, commonly adopted in theoretical analysis (Ratner et al., 2016), does not contradict our observation of high rank correlation between surrogate metrics.

The Borda method weighs all surrogate metrics equally. However, our experience and theoretical analysis suggest that focusing only on "good" metrics can improve performance after aggregation (see Sec. 5 and Corollary 1). But how do we identify "good" ranks without access to anomaly labels?

## 4.2 EMPIRICAL INFLUENCE AND ROBUST RANK AGGREGATION

Since we do not have access to anomaly labels, we introduce an intuitive unsupervised way to identify "good" or "reliable" rankings based on the notion of empirical influence. Empirical influence functions have been previously used to detect *influential* cases in regression (Cook & Weisberg, 1980). Given a collection of $M$ rankings $\mathcal{S} = \{\sigma_1, \cdots, \sigma_M\}$, let $\text{Borda}(\mathcal{S})$ denote the aggregate ranking from the Borda method. Then for some $\sigma_i \in \mathcal{S}$, empirical influence $\text{EI}(\sigma_i, \mathcal{S})$ measures the impact of $\sigma_i$ on the Borda solution.

$$\text{EI}(\sigma_i, \mathcal{S}) = f(\mathcal{S}) - f(\mathcal{S} \setminus \sigma_i), \quad \text{where } f(\mathcal{A}) = \frac{1}{|\mathcal{A}|} \sum_{i=1}^{|\mathcal{A}|} d_\tau(\text{Borda}(\mathcal{A}), \sigma_i) \qquad (2)$$

Under the assumption that a majority of rankings are good, EI is likely to identify bad rankings as ones with high positive EI. This is intuitive since removing a bad ranking results in larger decrease in the objective $f(\mathcal{S} \setminus \{\sigma_i\})$. In appendix A.9, we show that under idealized settings, empirical influence can identify bad rankings.

We cluster all rankings based on their EI using a two-component single-linkage agglomerative clustering (Murtagh & Contreras, 2012), discard all rankings from the "bad" cluster, and apply Borda aggregation to the remaining ones. To further improve ranking performance of aggregated rank, especially at the top, we only consider models at the top-$k$ ranks, setting the positions of the rest of the models to $N$ (Fagin et al., 2003). See Appendix A.7.1 for details. We collectively refer to these variations as Robust Borda rank aggregation.

## 5 EVALUATION RESULTS

### 5.1 DATASETS

We carry out experiments on two popular and widely used real-world collections with diverse time-series and anomalies:

**UCR Anomaly Archive (`UCR`) (Wu & Keogh, 2021)**   Wu and Keogh recently found flaws in many commonly used benchmarks for anomaly detection tasks. They introduced the UCR time-series Anomaly Archive, a collection of 250 diverse univariate time-series of varying lengths spanning domains such as medicine, sports, entomology, and space science, with natural as well as realistic synthetic anomalies overcoming the pitfalls of previous benchmarks. Given the heterogeneity of included time-series, we split UCR into 9 subsets by domain, (1) Acceleration, (2) Air Temperature, (3) Arterial Blood Pressure, ABP, (4) Electrical Penetration Graph, EPG, (5) Electrocardiogram, ECG, (6) Gait, (7) NASA, (8) Power Demand, and (9) Respiration, RESP. We will refer to each of these subsets as a separate *dataset*.

**Server Machine Dataset (`SMD`) (Su et al., 2019)**   SMD comprises of multivariate time-series with 38 features from 28 server machines collected from large internet company over a period of 5 weeks. Each entity includes common metrics such as CPU load, memory and network usage etc. for a server machine. Both train and test sets contain around 50,000 timestamps with 5% anomalous cases. While there has been some criticism of this dataset (Wu & Keogh, 2021), we use SMD due to its multivariate nature and the variety it brings to our evaluation along with UCR. We emphasize that on visually inspecting the SMD, we found most of the anomaly labels to be accurate, much like (Challu et al., 2022).

We refer to SMD as a single dataset, and thus in our evaluation consider 10 datasets (9 from UCR + SMD).

## 5.2 Model Set

There are over a hundred unique time-series anomaly detection algorithms developed using a wide variety of approaches. For our experiments, we implemented a representative subset of 5 popular methods (Table 1), across two different learning types (unsupervised and semi-supervised) and three different method families (forecasting, reconstruction and distance-based) (Schmidl et al., 2022). We defer brief descriptions of models to Appendix A.6.

We created a pool of **3** $k$-NN (Ramaswamy et al., 2000), **4** moving average, **4** DGHL (Challu et al., 2022), **4** LSTM-VAE (Park et al., 2018) and **4** RNN (Chang et al., 2017) models by varying hyper-parameters for each model, for a total **19** combinations. Finally, to efficiently train our models, we sub-sampled all time-series with $T > 2560$ by a factor of 10.

| Model | Area | Learning Type | Method Family |
|---|---|---|---|
| Moving Average | Statistics | Unsupervised | Forecasting |
| $k$-NN (Ramaswamy et al., 2000) | Classical ML | Semi-supervised | Distance |
| Dilated RNN (Chang et al., 2017) | Deep Learning | Semi-supervised | Forecasting |
| DGHL (Challu et al., 2022) | Deep Learning | Semi-supervised | Reconstruction |
| LSTM-VAE (Park et al., 2018) | Deep Learning | Semi-supervised | Reconstruction |

Table 1: *Models set and their properties*. Detailed descriptions of model architectures and hyper-parameters can be found in Appendix A.6.

## 5.3 Experimental Setup

**Evaluating model selection performance.** We evaluate all model selection strategies based on the adjusted best $F_1$ (Section 2) of the top-1 selected model. While there exist other popular measures to evaluate ranking algorithms such as Normalized Discounted Cumulative Gain (Wang et al., 2013), Mean Reciprocal Rank, etc., our evaluation metric is grounded in practice, since in our setting only one model will be selected and used.

Our approach can be used to select the best model for an individual time-series. Since anomalies are rare, a single time-series may contain only a few anomalies (e.g., one point anomaly in the case of some UCR time-series). This makes the *evaluation* of a selection strategy very noisy. To combat this issue, we perform model selection at the dataset level, i.e., we select a single method for a given dataset.

**Baselines.** As a baseline, we consider an approach in which we label a part of a dataset and select the best method based on these labels. We call this baseline "*supervised selection*" or SS in short. This represents current practice, where a subset of data is labeled, and the best performing model is used in the rest of the dataset. Specifically, we randomly partition each dataset into *selection* (20%) and *evaluation* (80%) sets. Each time-series in the dataset is assigned to one of these sets. The baseline method selects a model based on anomaly labels in the selection set. Our surrogate metrics do not use the selection set. Instead, the surrogate metrics are computed over the evaluation set without using the labels. For reference, we also report the expected performance when selecting a model from the model set at random ("random") and the performance of the actual best model ("oracle").

Adjusted best $F_1$ scores of the selected model (either by the baseline, surrogate metric or rank aggregation) are then reported. The entire experiment is repeated 5 times with random selection/evaluation splits.

**Model selection strategies.** We compare **17** individual surrogate metrics, including **5** prediction error metrics, **9** synthetic anomaly injection metrics, **3** metrics of centrality (average distance to 1, 3 and 5 nearest neighbors) (Section3); and **4** proposed robust variations of Borda rank aggregation (see Appendix A.7.2).

In the preliminary round of evaluations we found 5 of the anomaly injection metrics (*scale*, *noise*, *cutoff*, *contextual* and *speedup*) to generally outperform other surrogate metrics across all datasets (Appendix A.3). We apply rank aggregation only to these 5 metrics. While this subset of surrogate metrics was identified using labels, it does not appear to depend on the dataset. Thus, given a new dataset without labels, we still propose to use rank aggregation over the same 5 surrogate metrics.

**Pairwise-statistical tests.** We carry out one-sided paired Wilcoxon signed rank tests at a significance level of $\alpha = 0.05$ to identify significant performance differences between *all pairs* of model selection strategies and baselines on each dataset.

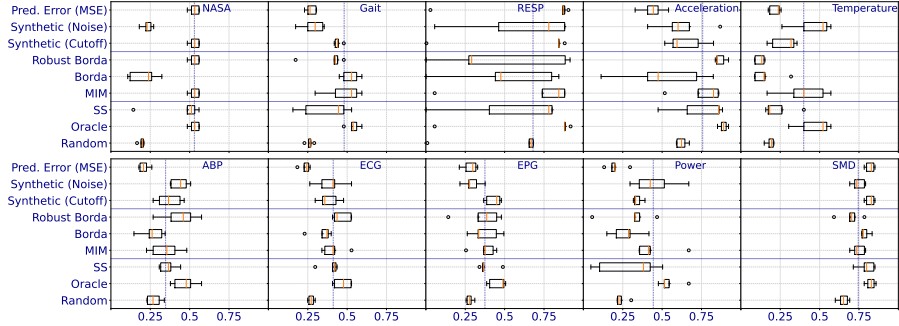

Figure 3: *Performance of the top-1 selected model.* Box plots of adjusted best $F_1$ of the model *selected* by each metric or baseline across 5 unique combinations of the selection and evaluation sets. Orange (|) and blue (|) vertical lines represent the median and average performance of each metric or baseline, and the minimum influence metric (MIM), respectively. Each box plot is organized into 3 sections: performance of individual metrics, Borda and its robust variations, and three baselines.

**Experimentation environment.**    All models and model selection algorithms were trained and built using `scikit-learn` 1.1.1 (Pedregosa et al., 2011), PyTorch 1.11.0 Paszke et al. (2019) along with Python 3.9.13. All our experiments were carried out on an AWS `g3.4xlarge` EC2 instance with with 16 Intel(R) Xeon(R) CPUs, 122 GiB RAM and a 8 GiB GPU.

## 5.4    RESULTS AND DISCUSSION

A subset of our results are shown in Figure 3 and Table 2. Here we provide results for model selection using some of the individual surrogate metrics, as well as three variations of rank aggregation: Borda, robust Borda and a special case of robust Borda where we select only one metric, the one that minimizes empirical influence, "*minimum influence metric*" or MIM (Section 4). The complete set of results with all ranking metrics can be found in Appendix A.3.

**No single best surrogate metric.**    Our results show that there is no surrogate metric which consistently selects the best model on all datasets. For instance, as can be seen in Figure 3, model selection based on prediction error (MSE) performs almost on par with the oracle on NASA, respiration (RESP) and server machine dataset (SMD), but worse than random model selection on the electrocardiogram (ECG), Arterial Blood Pressure (ABP) and acceleration datasets.

**Minimum Influence Metric perform on par with Supervised Selection.**    MIM is never significantly worse than SS and is in fact better on the air temperature (Temperature) dataset. Robust Borda on the other hand, is on par with SS on **8** datasets and significantly worse only on Temperature and SMD.

| Ranking Methods/Baselines | Significant Wins | | Significant Losses | | |
| --- | --- | --- | --- | --- | --- |
| | SS | Random | Oracle | SS | Random |
| Pred.  Error (MSE) | 1 | 3 | 8 | 4 | 4 |
| Syn.   (Noise) | **2** | 5 | 6 | 0 | 0 |
| Syn.   (Cutoff) | 1 | **8** | 8 | 0 | 0 |
| Syn.   (Scale) | 1 | **8** | 7 | 0 | 0 |
| Syn.   (Contextual) | 1 | 5 | 7 | 0 | 0 |
| Syn.   (Average) | 0 | 6 | 7 | 1 | 0 |
| Robust Borda | 0 | 4 | **6** | 2 | 0 |
| Borda | 0 | 4 | 9 | 3 | 0 |
| MIM | 1 | **7** | **6** | 0 | 0 |
| SS | n/a | 4 | 7 | n/a | 0 |
| Oracle | 7 | 10 | n/a | 0 | 0 |
| Random | 0 | n/a | 10 | 4 | n/a |

Table 2: *Results of pairwise-statistical tests.* On a total of $n = 10$ datasets, minimum influence metric (MIM) and robust Borda have the fewest significant losses to oracle, and are considerably better than random model selection and Borda.

**Robust variations of Borda outperform Borda.**    Aggregating via robust Borda and MIM are significantly better than Borda aggregation on **4** and **3** datasets, respectively. While MIM is never significantly worse than Borda, Robust Borda is inferior to Borda on SMD.

**Robust variations of Borda minimizes losses to Oracle.**    From Table 2 it is evident that MIM and Robust borda have fewer significant losses to the oracle in comparison to the 5 surrogate metrics used as their input, and compared to SS. Thus, in general, robust aggregation is better than any randomly chosen individual surrogate metric. In terms of significant improvements over supervised and random selection, we found MIM to be a better alternative than Robust Borda.

**Synthetic anomaly injection metrics outperform other classes of metrics.** We found synthetic anomaly injection to perform better than prediction error and centrality metrics (Appendix A.3). Among the three surrogate classes, centrality metrics performed the worst, most likely due to highly correlated bad rankings.

**Prior knowledge of anomalies might help identify good anomaly injection metrics.** For instance, synthetic metrics such as *speedup* and *cutoff* perform well on respiration data since these datasets are likely to have such anomalies. The slowing down of respiration rate is a common anomaly and indicative of sleep apnea[2]. Similarly, synthetic noise performs well on air temperature dataset, likely because time-series have noise anomalies to simulate the sudden failure of temperature sensors. We also hypothesize that the superior performance of synthetic noise might be due to the intrinsic noise and distortion added to $100$ of the $250$ UCR datasets (Wu & Keogh, 2021), thereby allowing the metric to choose models which are better suited to handle the noise. In practice, domain experts are aware of the type of anomalies that might exist in the data and using our anomaly injection methods; this prior knowledge can be effectively leveraged for model selection.

## 6 RELATED WORK

**Time-series Anomaly Detection.** The field has a long and rich history with well over a hundred proposed algorithms, differing in scope (e.g., multivariate vs univariate, unsupervised vs supervised), detection approach (e.g., forecasting, reconstruction, tree-based, etc.), etc. However, recent surveys have found that there is no single universally best method and the choice of the best method depends on the dataset (Schmidl et al., 2022; Blázquez-García et al., 2021).

**Meta Learning and Model Selection.** These approaches aim to identify the best model for a given dataset based on its characteristics such as the number of classes, attributes, instances etc. Recently, Zhao et al. (2021) explored the use of meta-learning to identify the best outlier detection algorithm on tabular datasets. For time-series datasets, both Ying et al. (2020) and Zhang et al. (2021) also approached model selection as meta-learning problem. All these studies rely on historical performance of models on a subset of "meta-train" datasets to essentially learn a mapping from the characteristics of a dataset ("meta-features") to model performance. These methods fail if anomaly labels are unavailable for meta-train datasets, or if the meta-train datasets are not representative of test datasets. Both these assumptions are frequently violated in practice, thus limiting the utility of such methods. In our approach, we do not assume access to similar datasets with anomaly labels.

**Weak Supervision and Rank Aggregation.** Our work draws on intuition from a large body of work on learning from weak or noisy labels since predictions of each surrogate metric have inherent noise and biases. We refer the reader to recent surveys on programmatic weak supervision, e.g. (Zhang et al., 2022a). Since we are interested in ranking anomaly detection methods rather than inferring their exact performance numbers, rank aggregation is a much more relevant paradigm for our problem setting.

An extended version of Related Work is provided in Appendix A.1.

## 7 CONCLUSION

We consider the problem of unsupervised model selection for anomaly detection in time-series. Our model selection approach is based on surrogate metrics which correlate with model performance to varying degrees but do not require anomaly labels. We identify three classes of surrogate metrics namely, *prediction error*, *synthetic anomaly injection*, and *model centrality*. We devise effective procedures to inject different kinds of synthetic anomalies and extend the idea of model centrality to time-series. Next, we propose to combine rankings from different metrics using a robust rank aggregation approach and provide theoretical justification for rank aggregation. Our evaluation using over 5000 trained models and 17 surrogate metrics shows that our proposed approach performs on par with selection based on partial data labeling. Future work might focus on other definitions of model centrality and include more synthetic anomalies types.

---

[2]Sleep apnea is a common and potentially serious health condition in which breathing repeatedly stops and starts (https://www.nhlbi.nih.gov/health/sleep-apnea).

## REPRODUCIBILITY STATEMENT

We provide an open-source implementations of our model selection method and anomaly detection models at `https://github.com/mononitogoswami/tsad-model-selection`.

All datasets used in this work are available in the public domain. Directions to download the Server Machine Dataset are available at `https://github.com/NetManAIOps/OmniAnomaly`, whereas the UCR Anomaly Archive can be downloaded from `https://www.cs.ucr.edu/~eamonn/time_series_data_2018/`.

## ACKNOWLEDGMENTS

We would like to thank Abishek Sankararaman, Anoop Deoras, Barış Kurt, Christos Faloutsos, Kelvin Kan, and Peihong Jiang for valuable discussions about the problem statement and methods. We thank the anonymous reviewers and area chairs for helpful suggestions. The first author would also like to thank AWS for travel grant to attend the conference.

## ETHICS STATEMENT

We propose an unsupervised method for model selection of time-series anomaly detection models. In this paper, we restrict ourselves to finding accurate anomaly detection models. The fairness of models chosen via unsupervised model selection is an open problem. While prevailing wisdom suggests a tension between the accuracy and fairness of models, recent work suggests otherwise. In fact, Wick et al. (2019) argue that under reasonable assumptions and certain definitions of fairness, accuracy and fairness are no longer in tension when selection and label bias are accounted for. We believe that an interesting direction of future work is extending our model selection problem and method to account for possibly competing objectives such as accuracy, fairness, inference time etc.

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

# A APPENDIX

## A.1 RELATED WORK

**Time-series Anomaly Detection.** This field has a long and rich history with well over a hundred proposed algorithms (Schmidl et al. (2022); Blázquez-García et al. (2021)). The methods vary in their scope (e.g., multivariate vs univariate, unsupervised vs supervised), and in the anomaly detection approach (e.g., forecasting, reconstruction, tree-based, etc.). At a high level many of these methods learn a model for the "normal" time-series regime, and identify points unlikely under the given model as anomalies. For example, progression of a time-series can be described using a forecasting model, and points deviating from the forecast can be flagged as outliers (Malhotra et al. (2015)). In another example, for tree-based methods, a data model is an ensemble of decision trees that recursively partition the points (sub-sequences of the time-series), until each sample is isolated. Abnormal points are easier to separate from the rest of the data, and therefore these will tend to have shorter distances from the root (Guha et al. (2016)). For a comprehensive overview of the field, we refer the reader to one of the recent surveys on anomaly detection methods and their evaluation (Schmidl et al. (2022); Blázquez-García et al. (2021); Wu & Keogh (2021)). Importantly, there is no universally best method, and the choice of the best method depends on the data at hand (Schmidl et al. (2022)).

**Meta Learning and Model Selection.** These approaches aim to identify the best model for the given dataset. The use of meta learning for model selection is a long-standing idea. For instance, Kalousis and Hilario used boosted decision trees to predict (propose) classifiers for a new dataset given data characteristics "meta-features") such as the number of classes, attributes, instances etc (Alexandros & Melanie, 2001)). More recently, Zhao et al. (2021) explored the use of meta-learning for outlier detection algorithms. Their algorithm relies on a collection of historical outlier detection datasets with ground-truth (anomaly) labels and historical performance of models on these meta-train datasets. The authors leverage meta-features of datasets to build a model that "recommends" an anomaly detection method for a dataset. Concurrently, Kotlar et al. (2021) developed a meta-learning algorithm with novel meta-features to select anomaly detection algorithms using labeled data in the training phase.

Ying et al. (2020) focused on model selection at the level of time-series. They characterize each time-series based on a fixed set of features, and use anomaly-labeled time-series from a knowledge base to train a classifier for model selection and a regressor for hyper-parameter estimation. A similar approach is taken by Zhang et al. (2021) who extracted time-series features, identified best performed models via exhaustive hyper-parameter tuning, and trained a classifier (or regressor) using the extracted time-series features and information about model performance as labels.

In contrast to methods mentioned in this sub-section, we are interested in unsupervised model selection. In particular, given an unlabeled dataset, we do not assume access to a similar dataset with anomaly labels.

**Weak Supervision and Rank Aggregation.** We consider a number of unsupervised metrics, and these can be viewed as weak labels or noisy human labelers. The question is then which metric to use or how to aggregate the metrics. There has been a substantial body of work on reconciling weak labels, mostly in the context of classification, and to a lesser extend for regression. Many such methods build a joint generative model of unlabeled data and weak labels to infer latent true labels. We refer the interested reader to a recent survey by Zhang et al. (2022a).

Since we are interested in ranking anomaly detection methods rather than inferring their exact performance numbers, rank aggregation is a more relevant paradigm for our project. Rank aggregation concerns with producing a consensus ranking given a number of potentially inconsistent rankings (permutations). One of the dominant approaches in the field involves defining a (pseudo-)distance in the space of permutations and then finding a barycentric ranking, i.e., the one that minimizes distance to other rankings (Korba et al. (2017)). The distance distribution can then be modeled, e.g., using the Mallows model, and rank aggregation reliability can be added by assuming a mixture from Mallows distributions with different dispersion parameters (Collas & Irurozki (2021)). In our work, we draw on the ideas from this research.

**Automatic Machine Learning (AutoML).**    The success machine learning across a gamut of applications has led to an ever-increasing demand of predictive systems that can be used by non-experts. However, current AutoML systems are either limited to classification with labeled data (e.g., see AutoML and **C**ombined **A**lgorithm **S**election and **H**yperparameter Optimization (CASH) problems in (Feurer et al., 2015)), or self-supervised regression or forecasting problems (Feurer et al., 2015; Gijsbers et al., 2022; 2019).

**Using Forecasting Error to Select Good Forecasting Models.**    Some prior work on anomaly detection time-series anomaly detection use the idea of selecting models with low forecasting error to pick the best model for forecasting-based anomaly detection (Saganowski & Andrysiak, 2020; Laptev et al., 2015; Kuang et al., 2022).

**Semi-supervised anomaly detection model selection.**    Recently, Zhang et al. (2022b) leveraged reinforcement learning to select the best base anomaly detection model given a dataset. They too use the idea of model centrality (referred to as Prediction-Consensus Confidence). However, their method is semi-supervised since it relies on actual anomaly labels to evaluate the reward function. Moreover, the base models do not include any recent deep learning-based anomaly detection models. Finally, they evaluate their method on only one dataset (SWaT).

**Limitations of Prior Work in the Context of Unsupervised Time-series Anomaly Detection Model Selection.**    In this section we briefly summarize the limitations of prior work considering the model selection problem in different settings.

- **Limited, predefined model set** $\mathcal{M}$: Many studies experiment with a small number of models for experiments ($|\mathcal{M}| \leq 5$). Most of these models are not state-of-the-art for time-series anomaly detection (e.g. Saganowski & Andrysiak (2020) use Holt-winters and ARFIMA), unrepresentative (only forecasting-based non-deep learning models), and specialised to certain domains (e.g. network anomaly detection (Saganowski & Andrysiak, 2020; Kuang et al., 2022)). In contrast, our experiments are conducted with **19** models of **5** different types (distance, forecasting and reconstruction-based), which have been shown to perform well for general time-series anomaly detection problems (Schmidl et al., 2022; Paparrizos et al., 2022b). Moreover, our model selection methodology is agnostic to the model set. Note that neither the model centrality and synthetic anomaly injection metrics, nor robust rank aggregation relies on the nature or hypothesis space of the models.

- **Limited, specialised datasets or old benchmarks with known issues**: Most studies (Saganowski & Andrysiak, 2020; Kuang et al., 2022; Laptev et al., 2015) are evaluated on a limited number of real-world datasets ($N <= 2$), of specific domains (e.g. network anomaly detection (Saganowski & Andrysiak, 2020; Kuang et al., 2022)). Most papers are either not evaluated on common anomaly detection benchmarks (Saganowski & Andrysiak, 2020; Kuang et al., 2022), or evaluated on old benchmarking datasets (e.g. Numenta Anomaly Benchmark) with known issues (Wu & Keogh, 2021). On the other hand, our methods are evaluated on **275** diverse time-series of different domains such as electrocardiography, air temperature etc.

- **Supervised model selection:** Some prior work, e.g. (Kuang et al., 2022) used supervised model selection methods such as Bayesian Information Criterion (BIC). However, in our setting methods such as BIC cannot be used because we do not have anomaly labels and measuring model complexity of different model types (deep neural networks, non-parametric models etc.) in our case is non-trivial.

- **No publicly available code**: Finally, most papers do not have publicly available implementations.

## A.2 Limitations and Future Work

**Computational Complexity.** Our methods are computationally expensive since they rely on predictions from all models in the model set. However, our main baseline is supervised selection which requires manually labeled data. Our premise is that manual labeling is much more expensive than compute time. For example, Amazon Web Services compute time in the order of cents/hour, where human time cost approximately $10/hour. In fact, recall that some of our benchmarks involve medical datasets where gold standard expert annotations can cost anywhere between $50-200 per hour (Abend et al., 2015).

**Assumption: A majority of rankings can identify accurate models better than random.** Empirical influence and robust borda methods provably identify good models only if a majority of the rankings can identify good models better than random (see appendices A.8, A.9). An interesting direction of future work is to relax this assumption, or make an orthogonal assumption. For example, there has been work on provably-optimal model selection of bandit algorithms which assume a specific structure of the problem (e.g., linear contextual bandits), and that at least one base model satisfies a regret guarantee with high probability (Pacchiano et al., 2020).

**Assumption: Partial orderings rather than total ordering.** We assume that all metrics impart a total ordering of all the models. However, this may not always be true. In our experience, some models may have the same performance as measured by a metric. To this end, future work may focus on aggregation techniques of partial rankings (Ailon, 2010).

**Online model selection.** We rely on predictions of models on the test data, however, in practice test data may be unavailable during model selection.

## A.3 Additional Results

The complete set of results are shown in Fig. 4 and Tab. 3.

We compare **17** individual surrogate metrics, including **5** prediction error metrics, **9** synthetic anomaly injection metrics, **3** metrics of centrality (average distance to 1, 3 and 5 nearest neighbors) (Section3); and **4** proposed robust variations of Borda rank aggregation, namely *Partial Borda*, *Trimmed Borda*, *Minimum Influence Metric* and *Robust Borda* (see Appendix A.7.2). We also compare Borda and its variations with exact Kemeny aggregation Appendix A.7.3).

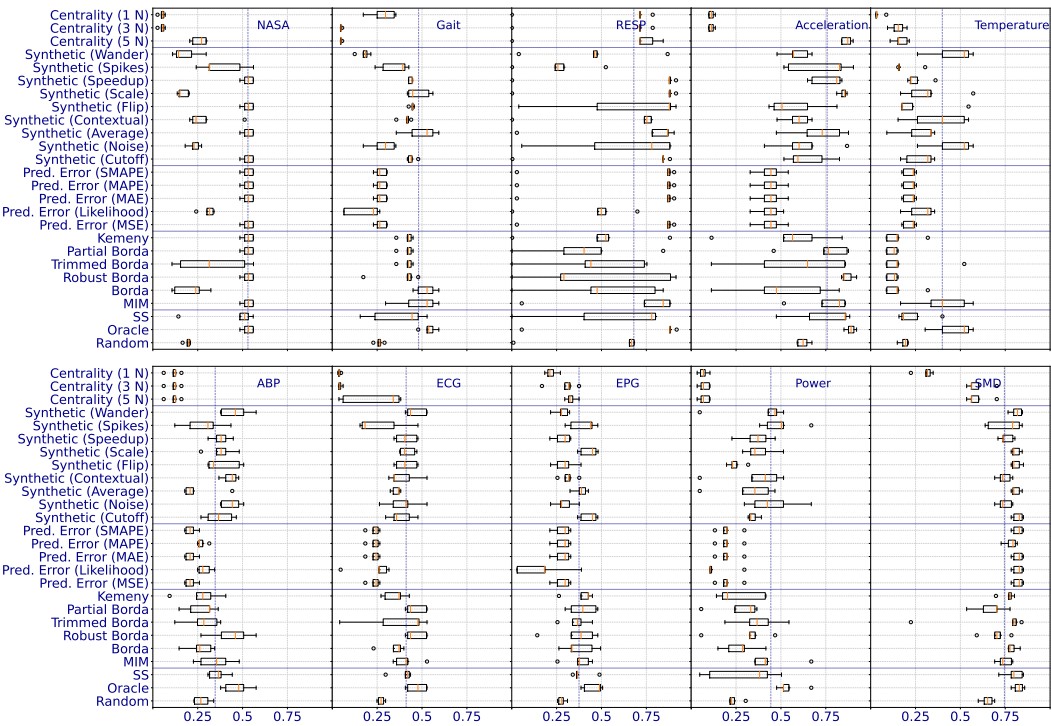

Figure 4: *Performance of the top-1 selected model.* Box plots of adjusted best $F_1$ of the model *selected* by each metric or baseline across 5 unique combinations of the selection and evaluation sets. Orange (|) and blue (|) vertical lines represent the median and average performance of each metric or baseline, and the minimum influence metric (MIM), respectively. Each box plot is organized into three sections: performance of individual metrics, Borda and its robust variations, and three baselines.

| Ranking Methods/Baselines | Significant Wins | | Significant Losses | | |
|---|---|---|---|---|---|
| | SS | Random | Oracle | SS | Random |
| Random | 0 | 0 | 10 | 4 | 0 |
| Oracle | 7 | 10 | 0 | 0 | 0 |
| SS | 0 | 4 | 7 | 0 | 0 |
| MIM | 1 | 7 | 6 | 0 | 0 |
| Borda | 0 | 3 | 9 | 3 | 0 |
| Robust Borda | 0 | 4 | 6 | 2 | 1 |
| Trimmed Borda | 0 | 1 | 8 | 0 | 0 |
| Partial Borda | 0 | 4 | 8 | 3 | 1 |
| Kemeny | 0 | 4 | 9 | 1 | 0 |
| Pred. Error (MSE) | 1 | 3 | 8 | 4 | 4 |
| Pred. Error (Likelihood) | 0 | 3 | 9 | 3 | 2 |
| Pred. Error (MAE) | 1 | 3 | 8 | 4 | 4 |
| Pred. Error (MAPE) | 1 | 3 | 9 | 4 | 3 |
| Pred. Error (SMAPE) | 1 | 3 | 8 | 4 | 4 |
| Synthetic (Cutoff) | 1 | 8 | 8 | 0 | 0 |
| Synthetic (Noise) | 2 | 5 | 6 | 0 | 0 |
| Synthetic (Average) | 0 | 6 | 7 | 1 | 0 |
| Synthetic (Contextual) | 1 | 5 | 7 | 0 | 0 |
| Synthetic (Flip) | 0 | 5 | 8 | 0 | 0 |
| Synthetic (Scale) | 1 | 8 | 7 | 0 | 0 |
| Synthetic (Speedup) | 1 | 6 | 9 | 1 | 0 |
| Synthetic (Spikes) | 0 | 3 | 8 | 1 | 1 |
| Synthetic (Wander) | 1 | 4 | 6 | 1 | 1 |
| Centrality (5 N) | 0 | 3 | 10 | 4 | 3 |
| Centrality (3 N) | 0 | 0 | 10 | 6 | 7 |
| Centrality (1 N) | 0 | 0 | 10 | 7 | 7 |

Table 3: *Results of pairwise-statistical tests..* On a total of $n = 10$ datasets, minimum influence metric (MIM) and robust Borda have the fewest significant losses to oracle, and are much better than random model selection and Borda.

## A.4 SYNTHETIC DATA EXPERIMENTS

The Mallows's model (Fligner & Verducci, 1986) is a popular distance-based distribution to model rank data. The probability mass function of a Mallows's model with central permutation $\sigma_0$ and dispersion parameter $\theta$ is given by:

$$p(\sigma) = \frac{exp(-\theta d(\sigma, \sigma_0))}{\psi}$$

where $d$ is the Kendall's $\tau$ distance. With increase in $\theta$, there is stronger consensus, thereby making Kemeny aggregation easier.

We aim to empirically answer two research questions:

1. What is the impact of noisy permutations on the median rank?
2. Can empirical influence identify outlying permutations?

To this end, we draw $k$ permutations from the Mallow's distribution $\sigma_1, \cdots \sigma_k \sim \mathcal{M}(\sigma_0, \theta)$ and $N - k$ permutations at uniformly at random from $S_n$. We compute the median rank $\hat{\sigma}$ using the Borda method. We experiment with $n \in \{50, 100\}$, $\theta \in \{0.05, \ 0.1, \ 0.2\}$ and $N = 50$ based on the experimental setup used by Ali & Meilă (2012).

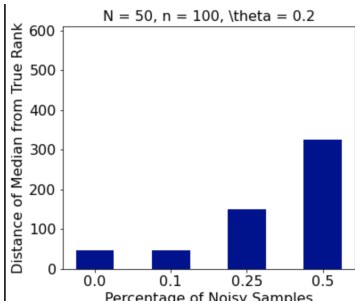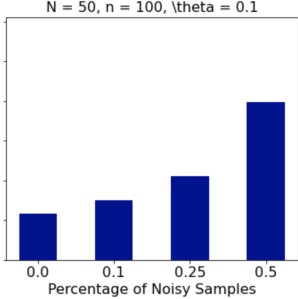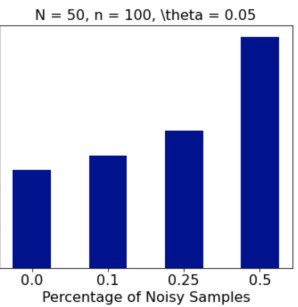

Figure 5: *Impact of Noisy Permutations.* With increase in noise, the distance of the median permutation ($\hat{\sigma}$) from the central permutation ($\sigma_0$) increase *i.e.* noise hurts rank aggregation.

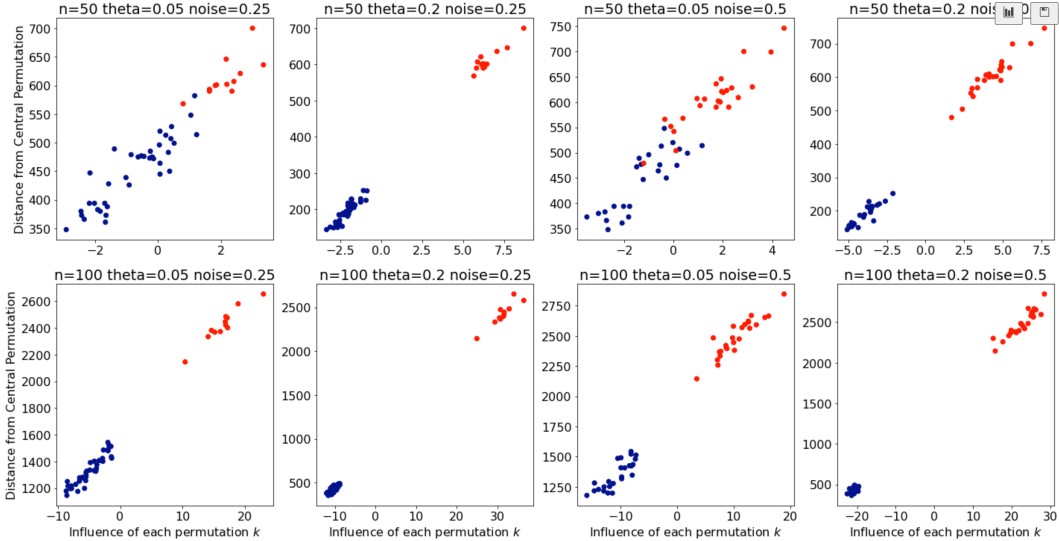

Figure 6: *Empirical Influence can Identify Outlying Permutations.* Influence of a permutation is directly proportional to its distance from the central permutation ($\sigma_0$). Under the Mallow's model, outlying (or low probability) permutations have a higher distance ($\sigma_0$).

## A.5 GENERATING SYNTHETIC ANOMALIES

Natural anomalies do not occur randomly, but are determined by the periodic nature of time-series. Therefore, before injecting anomalies we determine periodicity of time-series by finding the its auto-correlation, and only inject anomalies (barring spikes which are point-outliers) at beginning of a cycle. We only inject a single anomaly in each time-series in a repetition, and its length is chosen uniformly at random from [1, max_length], the latter being a user defined parameter (Fig. 7).

**Spike Anomaly.** To introduce a spike, we first draw $m \sim \texttt{Bernoulli}(p)$, and if $m = 1$ we inject a spike whose magnitude is governed by a normal distribution $s \sim \mathcal{N}(0, \sigma^2)$.

**Flip Anomaly.** We reverse the time-series:

```
ts_a[i, start:end] = ts[i, start:end][::-1]
```

**Speedup Anomaly.** We increase or decrease the frequency of a time-series using interpolation, where the factor of change in frequency ($\frac{1}{2}$ or 2) is a user-defined parameter.

**Noise Anomaly.** We inject noise drawn from a normal distribution $\mathcal{N}(0, \sigma^2)$, where $\sigma$ is a user-defined parameter.

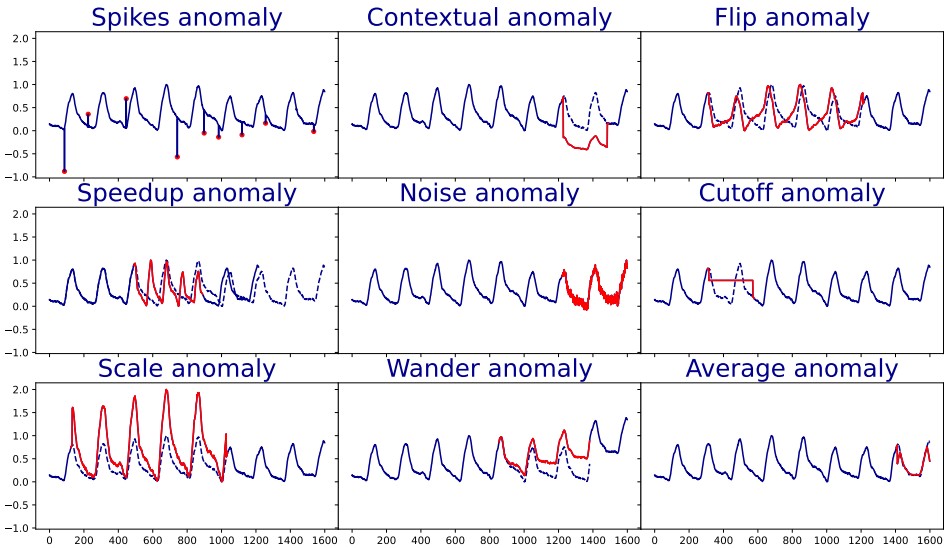

Figure 7: *Different kinds of synthetic anomalies.* We inject 9 different types of synthetic anomalies randomly, one at a time, across multiple copies of the original time-series. The - - is original time-series before anomalies are injected, – is the injected anomaly and the solid – is the time-series after anomaly injection.

**Cutoff Anomaly**   Here, we set the time-series to either $\mathcal{N}(0, \sigma^2)$ or $\mathcal{N}(1, \sigma^2)$, where both the choice of location parameter and $\sigma$ is user-defined.

**Average Anomaly.**   We run a moving average where the length of the window (`len_window`) is determined by the user:

```
ts_a[i, start:end] = moving_average(ts[i, start:end], len_window)
```

**Scale Anomaly.**   We scale the time-series by a user-defined factor (`factor`):

```
ts_a[i, start:end] = factor*ts[i, start:end]
```

**Wander Anomaly.**   We add a linear trend with the increase in baseline (`baseline`) defined by users:

```
ts_a[i, start:end] = np.linspace(0, baseline, end-start) + ts[i, start:end]
```

**Contextual Anomaly.**   We convolve a the time-series subsequence linear function $aY + b$, where $a \sim \mathcal{N}(1, \sigma_a^2)$ and $b \sim \mathcal{N}(0, \sigma_b^2)$

A.6   DESCRIPTION OF MODELS

There are over a hundred unique time-series anomaly detection algorithms developed using a wide variety of approaches. For our experiments, we implemented a representative subset of 5 popular methods (Table 1), across two different learning types (unsupervised and semi-supervised) and three different method families (forecasting, reconstruction and distance-based) (Schmidl et al., 2022).

In the context of time-series anomaly detection, "semi-supervised" means requiring a part of time-series known to be anomaly-free. Each time-series in the UCR and SMD collections comprises of a train part with no anomalies and a test part with one (UCR) or more anomalies (SMD). We used the train part for training semi-supervised models.

We emphasize that our surrogate metrics do not rely on availability of the "clean" train part (and we didn't use the train part for evaluation). Moreover, while semi-supervised models are *supposed* to be trained on anomaly-free part, in practice they can still work (perhaps sub-optimally) after training on time-series that potentially contained anomalies.

The models we implemented can also be categorized as forecasting, reconstruction and distance methods. Forecasting methods learn to forecast the next few time steps based on the current context window, whereas reconstruction methods encode observed sub-sequences into a low dimensional latent space and reconstruct them from the latent space. In both cases, the predictions (forecasts or reconstruction) of the models are compared with the observed values. Finally, distance based models used distance metrics (e.g. Euclidean distance) to compare sub-sequences, and expect anomalous sub-sequences to have large distances to sub-sequences they are most similar to.

We created a pool of **3** $k$-NN (Ramaswamy et al., 2000), and **4** moving average, **4** DGHL (Challu et al., 2022), **4** LSTM-VAE (Park et al., 2018) and **4** RNN (Chang et al., 2017) by varying important hyper-parameters, for a total **19** models. Finally, to efficiently train our models, we sub-sampled all time-series with $T > 2560$ by a factor of 10.

| Model | Area | Learning Type | Method Family |
|---|---|---|---|
| Moving Average | Statistics | Unsupervised | Forecasting |
| $k$-NN (Ramaswamy et al., 2000) | Classical ML | Semi-supervised | Distance |
| Dilated RNN (Chang et al., 2017) | Deep Learning | Semi-supervised | Forecasting |
| DGHL (Challu et al., 2022) | Deep Learning | Semi-supervised | Reconstruction |
| LSTM-VAE (Park et al., 2018) | Deep Learning | Semi-supervised | Reconstruction |

Table 4: *Models in the model set and their properties.*

Below is brief description of each model.

**Moving Average.**   These models compare observations with the "moving" average of recent observations in a sliding window of predefined size. These methods assume that if the current observation differs significantly from moving average, then it must be an anomaly. Specifically, if $x_t$ denotes the current observation, $h$ represents the window size and let $\tau$ be an anomaly threshold, then $x_t$ is flagged as an anomaly if:

$$\left\| x_t - \frac{1}{h} \sum_{i=t-h-1}^{t-1} x_i \right\|_2^2 > \tau,$$

Moving average is one of the simplest, efficient yet practical anomaly detection models, frequently used by industry practitioners. Moreover, moving average is unsupervised since it does not require any normal training data.

$k$**-Nearest Neighbors ($k$-NN) (Ramaswamy et al., 2000).**   These distance-based anomaly detection models compare a window of observations to their $k$ nearest neighbour windows in the training data. If the current window is significantly far its $k$ closest windows on the training set, then it must be an outlier. In our implementation, the distance between time-series windows is measured using standard Euclidean distance. Specifically, let $w_t = \{x_{t-h}, \cdots, x_t\}$ denote the current window of size $h$, $w_1, \cdots, w_k$ be the $k$ nearest windows in the train set to $w_t$, $|| \cdot ||_2$ represent the euclidean

norm, and $\tau$ be an anomaly threshold. Then the current window is flagged as an anomaly if:

$$\frac{1}{k}\sum_{i=1}^{k}||\boldsymbol{w}_i - \boldsymbol{w}_t||_2^2 > \tau$$

**Dilated RNN (Chang et al., 2017).** Dilated Recurrent neural network (RNN) is a recently proposed improvement over vanilla RNNs, characterized by multi-resolution dilated recurrent skip connections. These skip connections alleviate problems arising from memorizing long term dependencies without forgetting short term dependencies, vanishing and exploding gradients, and the sequential nature of forward and back-propagation in vanilla RNNs. Given a historical window of time-series, we use a dilated RNN to forecast $T_{la}$ time steps ahead. Let $w_t = \{\boldsymbol{x}_t, \boldsymbol{x}_{T_{la}}\}$ denote the current window of observations, $\hat{\boldsymbol{w}}_t = \{\hat{\boldsymbol{x}}_t, \cdots, \hat{\boldsymbol{x}}_{T_{la}}\}$ be the forecasts from the dilated RNN model, and $\tau$ be an anomaly threshold. Then $\boldsymbol{x}_t$ is flagged as an anomaly if:

$$||\hat{\boldsymbol{w}}_t - \boldsymbol{w}_t||_2^2 > \tau$$

**DGHL (Challu et al., 2022).** This model is a recent state-of-the-art Generative model for time-series anomaly detection. It uses a Top-Down Convolution Network (CNN) to map a novel hierarchical latent representation to a multivariate time-series windows. The key difference with other generative models as GANs and VAEs is that is trained with the Alternating Back-Propagation algorithm. The architecture does not include an encoder, instead, latent vectors are directly sampled from the posterior distribution. Let $\boldsymbol{x}_t$ be the observation and $\hat{\boldsymbol{x}}_t$ the reconstruction at timestamp $t$, and $\tau$ be an anomaly threshold. Then $\boldsymbol{x}_t$ is flagged as an anomaly if:

$$||\hat{\boldsymbol{x}}_t - \boldsymbol{x}_t||_2^2 > \tau$$

**LSTM-VAE (Park et al., 2018).** The LSTM-VAE is a reconstruction based model that uses a Variational Autoencoder with LSTM components to model the probability of observing the target time-series. The model also uses the latent representation to dynamically adapt the threshold to flag anomalies based on the normal variability of the data. The anomaly scores corresponds to the negative log-likelihood of an observation, $\boldsymbol{x}_t$, based on the output distribution of the model, and it is flagged as an anomaly if the score is larger than a threshold $\tau$:

$$-\log p(\boldsymbol{x}_t) > \tau$$

## A.7 BORDA RANK AGGREGATION

Borda is a positional ranking system and is an unbiased estimator of the Kemeny ranking of a sample distributed according to the Mallows Model (Fligner & Verducci, 1988). Consider $N$ models and a collection of $M$ surrogate metrics $\mathcal{S} = \{\sigma_1, \cdots, \sigma_M\}$. We use $\sigma_k$ to both refer to the surrogate metric and the ranking it induces. Then under the Borda scheme, for any given metric $\sigma_k$, the $i^{th}$ model receives $N - \sigma_k(i)$ points. Thus, the $i^{th}$ model accrues a total of $\sum_{k=1}^{M}(N - \sigma_k(i))$ points. Finally, all the models are ranked in decreasing order of their total points and ties are broken uniformly at random. Borda can be computed in quasi-linear ($\mathcal{O}(M + N \log N)$) time.

### A.7.1 IMPROVING RANKING PERFORMANCE AT THE TOP

In model selection, we only care about top-ranked models. Thus, to improve model selection performance at the top-ranks, we only consider models at the top-$k$ ranks, setting the positions of the rest of the models to $N$ (Fagin et al., 2003). Here, $k$ is a user-specified hyper-parameter. Specifically, under the *top-k* Borda scheme, the $i^{th}$ model accrues a total of $\sum_{m=1}^{M} \mathbb{I}[\sigma_m(i) \leq k](N - \sigma_m(i))$ points. Intuitively, this increases the probability of models which have top-$k$ ranks consistently across surrogate metrics, to have top ranks upon aggregation.

### A.7.2 ROBUST VARIATIONS OF BORDA

We consider multiple variations of Borda, namely *Partial Borda*, *Trimmed Borda*, *Minimum Influence Metric* (`MIM`) and *Robust Borda*. We collectively refer to these as robust variations of Borda rank aggregation.

**Partial Borda** improves model selection performance, especially at the top ranks by only considering models at the top-$k$ ranks, as described in Appendix A.7.1.

**Trimmed Borda** is another robust variation of Borda, which only aggregates reliable ("good") permutations. Trimmed Borda identifies outlying or "bad" permutations as ones having high positive empirical influence.

**Minimum Influence Metric** (`MIM`) for a dataset is the surrogate metric with minimum influence. Recall that while high positive influence is indicative of "bad" permutations, low values of influence are indicative of "good" permutations.

**Robust Borda** performs rank aggregation only based on the top-$k$ ranks while truncating permutations with high positive influence. Hence, Partial Borda, Trimmed Borda and Minimum Influence Metric can be viewed as special cases of Robust Borda.

### A.7.3 EXACT KEMENY RANK AGGREGATION

The Kemeny rank aggregation problem can be viewed as a minimization problem on a weighted directed graph (Conitzer et al., 2006). Let $G = (V, E)$ denote a weighted directed graph with the models as vertices. For every pair of models, we define an edge $(i \rightarrow j) \in E$ and set its weight as $w_{ij} = |\sum_{k=1}^{N} \mathbb{I}_{i \succ_k j} - \sum_{k=1}^{N} \mathbb{I}_{j \succ_k i}|$. Here the indicator $\mathbb{I}_{i \succ_k j} = 1$ denotes that metric $k$ prefers model $i$ over model $j$, and hence $w_{ij}$ denotes the number of metrics which prefer model $i$ over $j$.

Then we can formulate the rank aggregation problem as a binary linear program:

$$
\begin{aligned}
\text{minimize} \quad & \sum_{e \in E} w_e x_e \\
\text{subject to} \quad & \forall i \neq j \in V, x_{ij} + x_{ji} = 1 \quad \text{(Anti-symmetry and totality constraints)} \\
& \forall i \neq j \neq k \in V, x_{ij} + x_{jk} + x_{ki} \geq 1 \quad \text{(Transitivity constraints)} \\
& \forall i \neq j, x_{ij} \in \{0, 1\} \quad \text{(Binary constraint)}
\end{aligned}
$$

Intuitively, we incur a cost for every metric's pairwise comparison that we do not honor.

A.8    THEORETICAL JUSTIFICATION FOR BORDA RANK AGGREGATION

Suppose that we have several surrogate metrics for ranking our anomaly detection models. Also suppose that each of these metrics is better than random ranking. Why would it be beneficial to aggregate the rankings induced by these metrics? Theorem 1 provides an insight into the benefit of Borda rank aggregation.

Suppose that we have $N$ anomaly detection models to rank. Consider two arbitrary models $i$ and $j$ with (unknown) ground truth rankings $\sigma^*(i)$ and $\sigma^*(j)$. Without the loss of generality, assume $\sigma^*(i) < \sigma^*(j)$, i.e., model $i$ is better.

Next, let's view each surrogate metric $k = 1, \ldots, M$ as a random variable $\Omega_k$ taking values in permutations $S_N$ (a set of possible permutations over $N$ items), such that $\Omega_k(i) \in [N]$ denotes the rank of item $i$. We are interested in properties of Borda rank aggregation applied on realizations of these random variables. Theorem 1 states that, under certain assumptions, the probability of Borda ranking making a mistake (i.e., placing model $i$ lower than $j$) is upper bounded in a way such that increasing $M$ decreases this bound.

**Theorem.** *Assume that $\Omega_k$'s are pairwise independent, and that $\mathbb{P}(\Omega_k(i) < \Omega_k(j)) > 0.5$ for any arbitrary models $i$ and $j$, i.e., each surrogate metric is better than random. Then the probability that Borda aggregation makes a mistake in ranking models $i$ and $j$ is at most $2 \exp\left(-\frac{M\epsilon^2}{2}\right)$, for some fixed $\epsilon$.*

*Proof.* We begin by defining random variables $X_k = 2\mathbb{I}[\Omega_k(i) < \Omega_k(j)] - 1$. Thus $X_k = 1$ if $\Omega_k(i) < \Omega_k(j)$ and $X_k = -1$, otherwise. Denote the expectation of these random variables as $E_k \triangleq \mathbb{E}[X_k]$. We assume that $E_k > \epsilon$, $\forall k \in [M]$ for some small $\epsilon > 0$, i.e., that each surrogate metric $k$ is better than random.

Next, define $S_M \triangleq \sum_{k=1}^{M} X_k$, $E_M = \sum_{k=1}^{M} E_k$. Note that if $S_M > 0$ then Borda aggregation will correctly rank $i$ higher than $j$.

If $\Omega_k$ are pairwise independent, then $X_k$ are pairwise independent, and we can apply Hoeffding's inequality for all $t > 0$:

$$\mathbb{P}(|S_M - E_M| \geq t) \leq 2 \exp(-\frac{2t^2}{4M})$$

$$\mathbb{P}(E_M - S_M \geq t) \leq 2 \exp(-\frac{t^2}{2M})$$

$$\mathbb{P}(S_M \leq 0) \leq 2 \exp(-\frac{(E_M)^2}{2M}), \quad \text{setting } t = E_M$$

$$\mathbb{P}(S_M \leq 0) \leq 2 \exp(-\frac{M\epsilon^2}{2}), \quad \text{because } \epsilon < E_k, \forall k$$

Recall, that $S_M < 0$ results in an incorrect ranking (ranking $j$ higher than $i$), and $S_M = 0$ results in a tie (that we break uniformly at random). Therefore, the probability that Borda aggregation makes a mistake in ranking items $i$ and $j$ is upper bounded as per the above statement.

$\square$

We do not assume that $\Omega_k$ are identically distributed, but we assume that $\Omega_k$'s are pairwise independent. We emphasize that the notion of independence of $\Omega_k$ as random variables is different from rank correlation between realizations of these variables (i.e., permutations). For example, two permutations drawn independently from a Mallow's distribution can have a high Kendall's $\tau$ coefficient (rank correlation). Therefore, our assumption of independence, commonly adopted in theoretical analysis (Ratner et al., 2016), does not contradict our observation of high rank correlation between surrogate metrics.

**Corollary 1.** *Under the same assumptions as Theorem 1, if Borda aggregation makes a mistake in ranking models $i$ and $j$ with probability at most $\delta > 0$, then $\min_{k} \mathbb{P}(\Omega_k(i) < \Omega_k(j)) > \sqrt{\frac{\log \frac{2}{\delta}}{2M}} + \frac{1}{2}$, where $k \in [M]$ and $i, j \in [N]$ are any two arbitrary models.*

*Proof.* The proof follows from upper bounding the probability that Borda makes a mistake in ranking models $i$ and $j$ by $\delta$:

$$\mathbb{P}(S_M \leq 0) \leq 2\exp(-\frac{M\epsilon^2}{2}) \leq \delta \implies \epsilon \geq \sqrt{\frac{2\log\frac{2}{\delta}}{M}} \tag{3}$$

Consider the following relationship between $\epsilon$ and $\min_k \mathbb{P}(\Omega_k(i) < \Omega_k(j)))$ for $k \in [M]$:

$$\epsilon \leq \min_k E_k = \min_k 2\mathbb{P}(\Omega_k(i) < \Omega_k(j)) - 1 = 2(\min_k \mathbb{P}(\Omega_k(i) < \Omega_k(j))) - 1 \tag{4}$$

From Equations 3 and 4, we can conclude that:

$$\min_k \mathbb{P}(\Omega_k(i) < \Omega_k(j)) \geq \sqrt{\frac{\log\frac{2}{\delta}}{2M}} + \frac{1}{2}$$

$\square$

To reduce chances of error, it follows from Corollary 1 that we can either increase the number of surrogate metrics, or collect a smaller number of highly accurate metrics. As an example, set $\delta = 0.05$ and consider $M = 20$ metrics. Then for the Borda to err at most $5\%$ of times, each surrogate metric must prefer $A_i$ over $A_j$ with probability at least $0.7$. This also highlights the importance of empirical influence in identifying and removing bad permutations from the Borda aggregation.

## A.9 THEORETICAL JUSTIFICATION FOR EMPIRICAL INFLUENCE

In this section we provide two arguments in support of Empirical Influence (EI). First, we note that in an idealized case, where all but one metrics are perfect, EI helps identify the imperfect metric.

**Lemma 2.** *Consider a set of $M$ surrogate metrics that consists of $(M-1)$ perfect metrics and one imperfect metric. The perfect metrics induce permutations $\sigma_1 = \cdots = \sigma_{M-1} = \sigma^*$, where $\sigma^*$ is the ground truth permutation, while the imperfect metric induces permutation $\sigma_M \neq \sigma^*$. Then $EI(\sigma_M) \geq EI(\sigma_i)$ for any $1 \geq i \geq M-1$.*

*Proof.* Let $\mathcal{S} = \{\sigma_1, \cdots, \sigma_M\}$, $f(\mathcal{A}) = \frac{1}{|\mathcal{A}|}\sum_{i=1}^{|\mathcal{A}|} d_\tau(\text{Borda}(\mathcal{A}), \sigma_i)$, and $1 \geq i \geq M-1$.

Then

$$\begin{aligned}\text{EI}(\sigma_M) - \text{EI}(\sigma_i) &= f(\mathcal{S}) - f(\mathcal{S} \setminus \sigma_M) - f(\mathcal{S}) + f(\mathcal{S} \setminus \sigma_i) \\ &= f(\mathcal{S} \setminus \sigma_i) - f(\mathcal{S} \setminus \sigma_M)\end{aligned}$$

Since, $\sigma_1 = \cdots = \sigma_{M-1} = \sigma^*$, we have $\text{Borda}(\mathcal{S} \setminus \sigma_M) = \sigma^*$ and $f(\mathcal{S} \setminus \sigma_M) = 0$.

Kendall tau distance $d_\tau$ is non-negative, hence $f(\mathcal{S} \setminus \sigma_i) \geq 0$.

We have that $\text{EI}(\sigma_M) \geq \text{EI}(\sigma_i)$. $\square$

Next, recall that $\text{Borda}(\mathcal{A})$ is an unbiased estimator of the central permutation in the Kemeny-Young problem. Given a set of permutations $\mathcal{A}$, the central permutation is defined as

$$c(\mathcal{A}) = \arg\min \frac{1}{|\mathcal{A}|}\sum_{\sigma \in \mathcal{A}}^{|\mathcal{A}|} d_\tau(\chi, \sigma), \tag{5}$$

where the minimization is over all possible permutations $\chi$ of $N$ items.

We now consider a modified notion of Empirical Influence that uses the central permutation instead of Borda approximation.

$$\text{EI}'(\sigma) = f'(\mathcal{S}) - f'(\mathcal{S} \setminus \sigma), \tag{6}$$

where $f'(\mathcal{A}) = \frac{1}{|\mathcal{A}|} \sum_{i=1}^{|\mathcal{A}|} d_\tau(c(\mathcal{A}), \sigma_i)$.

Using this modified definition, we show how EI is capable of distinguishing between permutations close and far from the central permutation. Recall that good permutations are those that are close to the central permutation.

**Lemma 3.** *Consider a set of permutations $\mathcal{S} = \{\sigma_1, \ldots, \sigma_{M-2}, \sigma_{ok}, \sigma_{bad}\}$, where $\sigma_{ok}$ and $\sigma_{bad}$ represent good and bad permutations, respectively.*

*Let $\sigma^*$ be the unknown ground truth permutation.*

*Next, for some $r > 0$, assume that $d_\tau(\sigma^*, \sigma_{ok}) < r$ and $d_\tau(\sigma^*, \sigma_{bad}) > 3r$.*

*We also assume that overall, our set $\mathcal{S}$ is of reasonable quality in the sense that $d_\tau(c(\mathcal{S} \setminus \sigma), \sigma^*) < r$ for any $\sigma \in \mathcal{S}$.*

*Then, we have that $EI'(\sigma_{bad}) > EI'(\sigma_{ok})$.*

*Proof.* Consider the difference

$$
\begin{aligned}
\text{EI}'(\sigma_{\text{bad}}) - \text{EI}'(\sigma_{\text{ok}}) &= f'(\mathcal{S}) - f'(\mathcal{S} \setminus \sigma_{\text{bad}}) - f'(\mathcal{S}) + f'(\mathcal{S} \setminus \sigma_{\text{ok}}) \\
&= f'(\mathcal{S} \setminus \sigma_{\text{ok}}) - f'(\mathcal{S} \setminus \sigma_{\text{bad}}) \\
&\propto \sum_{\sigma \in \mathcal{S} \setminus \sigma_{\text{ok}}} d_\tau(c_{\text{bad}}, \sigma) - \sum_{\sigma \in \mathcal{S} \setminus \sigma_{\text{bad}}} d_\tau(c_{\text{ok}}, \sigma)
\end{aligned}
$$

Here, $c_{\text{ok}} = c(\mathcal{S} \setminus \sigma_{\text{bad}})$ and $c_{\text{bad}} = c(\mathcal{S} \setminus \sigma_{\text{ok}})$.

We have that

$$
\sum_{\sigma \in \mathcal{S} \setminus \sigma_{\text{ok}}} d_\tau(c_{\text{bad}}, \sigma) - \sum_{\sigma \in \mathcal{S} \setminus \sigma_{\text{bad}}} d_\tau(c_{\text{ok}}, \sigma)
$$

$$
= \sum_{\sigma \in \mathcal{S} \setminus \sigma_{\text{bad}}} d_\tau(c_{\text{bad}}, \sigma) - \sum_{\sigma \in \mathcal{S} \setminus \sigma_{\text{bad}}} d_\tau(c_{\text{ok}}, \sigma) + d_\tau(c_{\text{bad}}, \sigma_{\text{bad}}) - d_\tau(c_{\text{bad}}, \sigma_{\text{ok}})
$$

Under our assumptions, $d_\tau(c_{\text{bad}}, \sigma^*) < r$, $d_\tau(\sigma_{\text{bad}}, \sigma^*) > 3r$ and $d_\tau(\sigma_{\text{ok}}, \sigma^*) < r$.

Therefore $\text{EI}'(\sigma_{\text{bad}}) > \text{EI}'(\sigma_{\text{ok}})$.

$\square$

## A.10 CORRELATION BETWEEN ADJUSTED BEST $F_1$ AND VUS-ROC

**Motivation.** Paparrizos et al. (2022a) in a recent study found the volume under the surface of the ROC curve (VUS-ROC) to be more robust than other compared evaluation metrics. Their study did not compare VUS-ROC against adjusted best $F_1$, used in this paper. The VUS-ROC metric is defined for univariate time-series, whereas one of the datasets we use, the Server Machine dataset (SMD), has multivariate time-series. Hence, our main results are still with respect to adjusted best $F_1$. However, we carry out an experiment to evaluate the correlation between VUS-ROC correlates with adjusted best $F_1$.

**Experimental setup.** For every time-series, we computed the spearman-$r$ and kendall-$\tau$ correlation between the performance of models as measured by VUS-ROC and adjusted best $F_1$. We carried out this experiment on 250 time-series from the UCR anomaly archive (Wu & Keogh, 2021). We measured the dispersion of an evaluation measure (VUS-ROC or adjusted best $F_1$) on a time-series as the interquartile range of performance values of all models in that time-series. The sliding window parameter that VUS-ROC relies on, was automatically set to the auto-correlation of time-series.

**Results.** Out of 250 time-series, only 23 time-series did not have statistically significant positive kendall-$\tau$ and spearman-$r$ correlation. All these time-series had low dispersion of the evaluation measures *i.e.* the difference in performance of most models trained on the time-series, measured in terms of VUS-ROC or adjusted best $F_1$, was less than 10%.

## A.11    CORRELATION BETWEEN ADJUSTED BEST $F_1$ AND ADJUSTED PR-AUC

On our datasets, performance of models measured using PR-AUC and adjusted best $F_1$ had statistically significant highly positive correlation (see Figure 8.

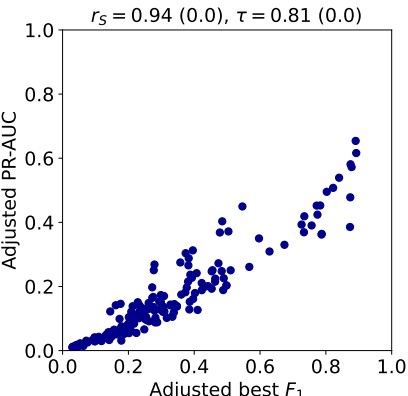

Figure 8: *Correlation between adjusted best $F_1$ and adjusted PR-AUC. $r_S$ and $\tau$ denote the spearman and kendall correlation, respectively. $p$-values of two-sided tests are reported in parenthesis.*

## A.12    DOES EMPIRICAL INFLUENCE IDENTIFY BAD METRICS?

To measure the correlation between empirical influence and quality of a metric, we provide (1) experiments on real-world datasets, and (2) an analysis on synthetic data in Appendix A.4, where we see that empirical influence can almost perfectly identify bad metrics.

**Experimental setup.**    We define the quality of a metric $\sigma$ using two quantities: ($Q1$) adjusted best $F_1$ of top-5 models ranked by a metric $\sigma$, ($Q2$) difference in the adjusted best $F_1$ of top-5 and bottom-5 models ranked by a metric $\sigma$. This quality metric is inspired by the separation analysis in Paparrizos et al. (2022a).

**Results.**    6 out of 9 datasets had negative kendall-$\tau$ correlation between the quality of a metric and its empirical influence, when quality is measured using $Q1$. Of these 6 datasets, 4 datasets had statistically significant negative correlation. Recall that bad metrics have high positive empirical influence (Section 4.2). Even when quality is measured using $Q2$, 6 out of 9 datasets had negative kendall-$\tau$ correlation between the quality of a metric and its empirical influence. Of these 6 datasets, 2 datasets had statistically significant negative correlation.

A.13 Results when aggregating over all metrics

When we perform robust rank aggregation over all the metrics, instead of only 5 anomaly injection metrics (scale, noise, cutoff, contextual and speedup), we observed a degradation in the performance of both minimum influence metric (MIM) and robust Borda. MIM has more significant wins and fewer significant losses when only the 5 best performing metrics are aggregated. This degradation in performance can be explained by the fact that some metrics have ranking performance worse than random model selection, for example model centrality metrics and prediction likelihood.

| Ranking Methods/Baselines | Significant Wins | | Significant Losses | | |
|---|---|---|---|---|---|
| | SS | Random | Oracle | SS | Random |
| MIM | 0 | 1 | 5 | 3 | 1 |
| Robust Borda | 0 | 2 | 6 | 2 | 1 |

Table 5: *Results of pairwise-statistical tests, when aggregation is performed using all metrics on $n = 6$ datasets.*

| Ranking Methods/Baselines | Significant Wins | | Significant Losses | | |
|---|---|---|---|---|---|
| | SS | Random | Oracle | SS | Random |
| MIM | 2 | 3 | 5 | 0 | 0 |
| Robust Borda | 2 | 4 | 5 | 2 | 1 |

Table 6: *Results of pairwise-statistical tests, when aggregation is performed using only best performing 5 metrics on $n = 6$ datasets.*

A.14 Does Theorem 1 hold in practice?

In this short experiment we provide empirical support for Theorem 1.

**Experimental setup.** Theorem 1 considers error bounds when ranking two models. Thus, for each time series we identify the best- and worst-performing models and empirically test whether aggregation over more rankings helps reduce this error. We varied the number of surrogate metrics from 2 to 17, and perform borda rank aggregation to find the best model. The performance of the aggregated borda rank is measured as the adjusted best $F_1$ of the top ranked model. For each combination of $m$ metrics, we report the performance averaged over all $\binom{17}{m}$ unique combinations of the metrics.

**Results.** First, we present the performance of the best model as identified by borda aggregation of a combination of metrics, averaged across all the time-series. Overall, we see a clear trend where the performance of the selected best model increases with the number of surrogate metrics (rankings).

| Average performance when using $m$ metrics | Adjusted best $F_1$ |
|---|---|
| $m = 2$ metrics | 0.588 |
| $m = 5$ metrics | 0.795 |
| $m = 8$ metrics | 0.784 |
| $m = 11$ metrics | 0.829 |
| $m = 14$ metrics | 0.821 |
| $m = 17$ metrics | 0.838 |

Table 7: *Adjusted best $F_1$ of the best model as identified by borda aggregation of all possible combinations of $m$ metrics, averaged across all time-series datasets. Increasing the number of surrogate metrics improves model selection performance.*

In only 27 out of 181 time-series, we did not observe a monotonic improvement as the number of metrics increased. However, we notice that in all such cases, the performance of all anomaly detection models was low and similar to each other. Under such circumstances, not only do surrogate metrics tend to be noisy, violating the assumptions of the theorem, but also, model selection does not matter much.

Below, as an example, we also report the results of two individual time-series where we can see our theorem at play, and another time-series where our theorem does not hold:

| Average performance when using $m$ metrics | Adjusted best $F_1$ |
|---|---|
| $m = 2$ metrics | 0.507 |
| $m = 5$ metrics | 0.701 |
| $m = 8$ metrics | 0.584 |
| $m = 11$ metrics | 0.742 |
| $m = 14$ metrics | 0.629 |
| $m = 17$ metrics | 0.984 |

Table 8: *Adjusted best $F_1$ of the best model as identified by borda aggregation of all possible combinations of $m$ metrics, on machine-3-6 time-series of the Server Machine dataset. Increasing the number of surrogate metrics improves model selection performance.*

| Average performance when using $m$ metrics | Adjusted best $F_1$ |
|---|---|
| $m = 2$ metrics | 0.566 |
| $m = 5$ metrics | 0.742 |
| $m = 8$ metrics | 0.645 |
| $m = 11$ metrics | 0.780 |
| $m = 14$ metrics | 0.677 |
| $m = 17$ metrics | 0.999 |

Table 9: *Adjusted best $F_1$ of the best model as identified by borda aggregation of all possible combinations of $m$ metrics, InternalBleeding17 time-series of the UCR Anomaly Archive. Increasing the number of surrogate metrics improves model selection performance.*

| Average performance when using $m$ metrics | Adjusted best $F_1$ |
|---|---|
| $m = 2$ metrics | 0.324 |
| $m = 5$ metrics | 0.396 |
| $m = 8$ metrics | 0.272 |
| $m = 11$ metrics | 0.268 |
| $m = 14$ metrics | 0.240 |
| $m = 17$ metrics | 0.240 |

Table 10: *Adjusted best $F_1$ of the best model as identified by borda aggregation of all possible combinations of $m$ metrics, on DISTORTEDInternalBleeding17 time-series of the UCR Anomaly Archive. In this time-series, increasing the number of surrogate metrics hurts model selection performance.*

