# OpenReview forum: "Unsupervised Model Selection for Time Series Anomaly Detection"
_ICLR.cc/2023/Conference — ICLR 2023 notable top 25%_

### Official Review · Reviewer_kY2N · 2022-10-21

**Confidence:** 4
**Correctness:** 3
**Technical Novelty And Significance:** 2
**Empirical Novelty And Significance:** 2
**Recommendation:** 5

**Clarity, Quality, Novelty And Reproducibility:**

Basically, I like the paper's motivation; it is quite a fundamental research problem to select the most accurate model for performing anomaly detection for time-series data. Besides, this paper is well-structured, and easy to follow. I appreciate that the paper conducted experiments based on many real datasets of various domains, such as medicine, sports, and entomology.

In terms of technical depth, the proposed approach seems to be a combination of the previous approach; it is technically somewhat shallow.

I am concerned about the computational cost of the proposed approach. It needs to compute several models, and the proposed approach itself has user-defined parameters needed to be tuned. The impact of the user-defined parameters for the proposed approach is unclear from the descriptions of the paper.

As shown in the experimental result, such as Figure 3 and 4, the effectiveness of the proposed approach is not high compared to the previous approaches, although it is not so worse than them. Since the proposed approach requires higher computational costs than the previous approach, it could not be so useful in performing anomaly detections.

**Strength And Weaknesses:**

Strength:
- Simple model to be easily implemented.
- General approach to accommodate various anomaly detection approaches.
- Extensive experiments are performed in the paper.

Weakness:
- The proposed approach is a combination of existing methods.
- Technical depth of the proposed approach is not so deep.
- The advantage of the proposed approach is not so significant compared to other approaches.

**Summary Of The Paper:**

This paper proposes an approach of combining unsupervised metrics to realize an effective selection of anomaly detection models for time-series data. The unsupervised are highly correlated with standard supervised anomaly detection performance metrics. Therefore, the proposed approach can detect anomalies in an unsupervised manner. Using datasets of various domains, the paper performed experiments to show the effectiveness of the proposed approach.

**Summary Of The Review:**

More theoretical analyses are required for the proposed approach.
The computational time of each approach should be shown in the paper.
The superiority of the proposed approach in terms of accuracy is not so high to accept the paper.

---

> ### Author Response · Authors · 2022-11-17
> **Summary of changes and responses**
>
> > *I am concerned about the computational cost of the proposed approach*
>
> > *the effectiveness of the proposed approach is not high, although it is not so worse than them*
>
> Our main baseline is supervised selection which requires manually labelled data. Our premise is that manual labelling is much more expensive than compute time. For example, Amazon Web Services compute time in the order of cents/hour, where human time cost approximately \\$10/hour. In fact, recall that some of our benchmarks involve medical datasets where gold standard expert annotations can cost anywhere between \\$50-200 per hour [1]. Accordingly, our main argument is that our method can achieve the same result as supervised selection, but without the need for manual labelling.
>
> We have now added a note about the computational complexity of our method in Appendix A.2 and have analyzed the complexity of rank aggregation in Appendix A.7.
>
> **References**
>
> [1] Nicholas S Abend, Alexis A Topjian, and Sankey Williams. How much does it cost to identify a critically ill child experiencing electrographic seizures? Journal of clinical neurophysiology: official publication of the American Electroencephalographic Society, 32(3):257, 2015.
>
> > *has user-defined parameters needed to be tuned*
>
> Our final recommendation is to use one of our methods minimum influence metric (MIM), which is the surrogate metric that minimises empirical influence. Therefore, there are no parameters. In fact, even robust Borda, another proposed method, does not have any hyper-parameter. Even hyper-parameters to evaluate synthetic anomaly injection and model centrality metrics are selected before-hand and do not need to be tuned.
>
> > *proposed approach seems to be a combination of the previous approach*
>
> The following contributions make our study non-trivial and novel:
> 1. Systematic exploration and evaluation of various surrogate metrics of anomaly detection model performance.
> 2. Public implementation and experimentation with a large number of intuitive anomaly injection metrics.
> 3. Adaptation of the idea of model centrality to time series anomaly detection, where the notion of centrality was not obvious.
> 4. Proposed robust rank aggregation strategy which performs on par with supervised selection of models based on manual labelling of data.
> 5. Analysis of the properties of borda rank aggregation (Theorem 1) for model selection
> 6. To the best of our knowledge, we propose one of the first methods for unsupervised selection of anomaly detection models on time-series data. Prior work has performed model selection in fields such as forecasting and network anomaly detection, but it is one of the first studies situated in time-series anomaly detection model selection in general settings.
> 7. We conduct experiments on over 275 diverse time-series, spanning a gamut of domains such as medicine, entomology, etc. with 5 popular and widely-used anomaly detection models, each with 1 to 4 hyper-parameter combinations, resulting in over 5,000 trained models.
> 8. Theoretical justification of why empirical influence identifies bad permutations.
>
> *In light of these changes and new results, we sincerely hope that the reviewer would consider increasing their score.*

---

### Official Review · Reviewer_E7ki · 2022-10-24

**Confidence:** 5
**Correctness:** 2
**Technical Novelty And Significance:** 2
**Empirical Novelty And Significance:** 2
**Recommendation:** 3

**Clarity, Quality, Novelty And Reproducibility:**

Clarity is reasonable. Novelty and quality are somewhat low, considering missed recent advances as well as many missed AutoML baselines.

**Strength And Weaknesses:**

Strengths

1. Time-series anomaly detection is a well-studied, timely, and important problem
2. Well-written and easy to follow
3. AutoML for anomaly detection for time series is not sufficiently studied and hence this study fills a gap

Weaknesses

1. Missing baselines from the AutoML area in general
2. Missing baselines from the AutoML area for time series
3. Omitted new benchmarks and evaluation strategies

Comments:

- AutoML is a very well-studied area. The paper claims it's the first work regarding time-series anomaly detection, which is also not true. A simple query reveals multiple solutions in that area that are unfortunately not mentioned and not compared against. It's very difficult to assess the importance of this work and if it advances state of the art without covering and comparing the relevant work appropriately

https://scholar.google.com/scholar?hl=en&as_sdt=0%2C36&q=model+selection+time-series+anomaly+detection&btnG=

Existing AutoML solutions for non-time-series data can very well be adopted for this problem and needs to be compared. Time-series specific AutoML solution for anomaly detection or other ML tasks should also be used in comparison.

- The work relies on specific recent papers to justify reasoning/choices (e.g., Schmidl et al.) while ignoring works appearing at the same time/conference, with different findings. Schmidl et al limited the time a method could run during their evaluation and used methods "as is" (i.e., not reimplemented), which resulted in model failures, which were not taken into consideration but instead decreased the overall performance of specific methods. This is truly problematic as their findings differ from other studies in this area, e.g., Paparrizos et al., which focused on significantly more data (12000+ time series) but less methods and identified several methods significantly outperforming current SOTA. Even though the claim that no method performs well across every single dataset still holds, several of the choices in the current papers are problematic (use of baselines, use of datasets, use of evaluation measures). Recent developments suggest using different evaluation measures in the first place, proving that the F measure is flawed.

"Tsb-uad: an end-to-end benchmark suite for univariate time-series anomaly detection." Proceedings of the VLDB Endowment 15.8 (2022): 1697-1711.

"Volume under the surface: a new accuracy evaluation measure for time-series anomaly detection." Proceedings of the VLDB Endowment 15.11 (2022): 2774-2787.









**Summary Of The Paper:**

The paper proposes a solution for selecting models for time-series anomaly detection. Starting from the assumption that no single model performs best across all datasets, the paper argues for the importance of AutoML solutions in that area. As a result, the paper first studies surrogate measures that correlate with the accuracy of anomaly detection methods. Then, it combines these measures using a rank aggregation technique. In the end, the paper demonstrates the effectiveness of the proposed solution.

**Summary Of The Review:**

See above. The missed AutoML baselines is a major flaw of this work. In addition, many recent advances in the time-series anomaly detection literature are not mentioned/ignored, which may change the outcome of this work.

---

> ### Author Response · Authors · 2022-11-17
> **Thematic analysis of related work (1/4)**
>
> > *The paper claims it's the first work regarding time-series anomaly detection …*
>
> > *A simple query reveals multiple solutions in that area …*
>
> Thank you for providing the query. Based on our review of query results, we do not believe that we have missed any important baselines. Overall, query results do not appear to contradict our statement that was specifically about unsupervised selection among anomaly detection models. Nevertheless, we have qualified our claim: “ *To the best of our knowledge, we propose one of the first methods for unsupervised selection of anomaly detection models on time-series data. To this end, we identify intuitive and effective unsupervised metrics for model performance. Prior work has used a few of these unsupervised metrics for problems other than time-series anomaly detection model selection.* ”
>
> Following is a thematic analysis of the top results as obtained by one of the authors using the query shared by the reviewer:
>
> ### Use forecasting error to select forecasting models
>
> Papers [1, 2, 3] use the idea of selecting among forecasting models before performing anomaly detection. However, these papers do not consider the following questions:
> Which forecasting quality metric should we use?
> How do we select an anomaly detection model not based on forecasting e.g. reconstruction, density or distance-based anomaly detection models? Note that a significant proportion of anomaly detection methods are not based on forecasting, as noted by Schmidl et al. [16] and Paparrizos et al. [17].
>
> None of these papers [1, 2, 3] have publicly available implementations. However, we did include forecasting performance as a surrogate metric in our evaluation. We have now added citations to these papers.
>
> **References**
>
> [1] Saganowski, Łukasz, and Tomasz Andrysiak. "Time series forecasting with model selection applied to anomaly detection in network traffic." Logic Journal of the IGPL 28.4 (2020): 531-545.
>
> [2] Laptev, Nikolay, Saeed Amizadeh, and Ian Flint. "Generic and scalable framework for automated time-series anomaly detection." Proceedings of the 21th ACM SIGKDD international conference on knowledge discovery and data mining. 2015.
>
> [3] Kuang, Ye, et al. "On the Modeling of RTT Time Series for Network Anomaly Detection." Security and Communication Networks 2022 (2022).
>
> ### Papers unrelated to anomaly detection model selection
>
> Papers [4, 5, 6] are not about anomaly detection model selection.
> In fact, the keyword “model selection” does not appear in [4].
>
> Choi et al. [5] highlight some best practices when choosing modelling approaches for an anomaly detection problem, but do not propose any model selection algorithm.
>
> Amarbayasgalan et al. [6] propose a time-series anomaly detection approach.
>
> **References**
>
> [4] Ren, Hansheng, et al. "Time-series anomaly detection service at microsoft." Proceedings of the 25th ACM SIGKDD international conference on knowledge discovery & data mining. 2019.
>
> [5] Choi, Kukjin, et al. "Deep learning for anomaly detection in time-series data: review, analysis, and guidelines." IEEE Access (2021).
>
> [6] Amarbayasgalan, Tsatsral, et al. "Unsupervised anomaly detection approach for time-series in multi-domains using deep reconstruction error." Symmetry 12.8 (2020): 1251.

---

> > ### Author Response · Authors · 2022-11-17
> > **Thematic analysis of related work (2/4)**
> >
> > ### Papers still under peer-review
> >
> > Papers [7, 8] appear to be arXiv only. They might have been not peer reviewed or reviewed and rejected. These papers have one or more of the following limitations:
> >
> > 1. **Limited, old benchmarking datasets with known issues**: These papers are evaluated on a limited number of benchmarking datasets with known issues. For instance, Jung et al. [7] report their results on one anomaly detection benchmark (Numenta Anomaly Benchmark) with known issues (see Wu et al. [18]).
> >
> > 2. **Evaluation on point-wise metrics**: As pointed by the reviewer, Paparrizos et al. [15] and our experience as noted in Section 2.1, pointwise metrics disregard the time-series nature of the problem. In fact, Jung et al. [7] fail to note whether they are measuring the area under the Precision-recall or the Receiver Operating Characteristics curve.
> >
> > 3. **Public code unavailable**: In addition to these shortcomings, their code is unavailable in the public domain.
> >
> > Jung et al. [7] present a new anomaly detection method, and not a model selection approach. The proposed anomaly detection method is an ensemble of models of a specific kind, but the heuristic used in model weighting is not generally applicable (e.g., a trivial model that does not detect anything would receive a high weight). Finally, evaluation results do not show advantages of the proposed model.
> >
> > Chatterjee et al. [8] use the idea of anomaly injection (scale, trend and spikes), which we have already included in the synthetic anomaly injection surrogate class. In fact, we include a richer variety of synthetic injection techniques e.g. speedup, cutoff etc. (see Appendix for the 9 different types of real-world anomalies that we include).
> >
> > **References**
> >
> > [7] Jung, Deokwoo, et al. "Time Series Anomaly Detection with label-free Model Selection." arXiv preprint arXiv:2106.07473 (2021).
> >
> > [8] Chatterjee, Sourav, et al. "MOSPAT: AutoML based Model Selection and Parameter Tuning for Time Series Anomaly Detection." arXiv preprint arXiv:2205.11755 (2022).
> >
> > ### Semi-supervised anomaly detection model selection
> >
> > Zhang et al. [9] leverage reinforcement learning to select the best base anomaly detection model given a dataset. They use the idea of model centrality (referred to as Prediction-Consensus Confidence), which we already included as one of surrogate metric  classes. However, their method is semi-supervised since it relies on actual anomaly labels to evaluate the reward function. Moreover, the base models do not include any recent deep learning-based anomaly detection models. Finally, they evaluate their method on only one dataset (SWaT).
> >
> > **References**
> >
> > [9] Zhang, Jiuqi Elise, Di Wu, and Benoit Boulet. "Time Series Anomaly Detection via Reinforcement Learning-Based Model Selection." arXiv preprint arXiv:2205.09884 (2022).

---

> > > ### Author Response · Authors · 2022-11-17
> > > **Regarding AutoML baselines (3/4)**
> > >
> > > > *Existing AutoML solutions for non-time-series data can very well be adopted for this problem and needs to be compared*
> > >
> > > > *Time-series specific AutoML solution for anomaly detection or other ML tasks should also be used in comparison.*
> > >
> > > We would like to stress that our focus is on unsupervised model selection. AutoML is predominantly used in a supervised learning setting (e.g., see CASH problem in Feurer et al. [10]  and Gijsbers et al. [11]). Moreover, AutoML for regression and time-series forecasting leverages the fact that a part of time series itself can be used as labels (e.g., see Auto Forecasting problem in Wang et al. [12]). In our work, we do consider forecasting performance as a class of surrogate metrics, but our ultimate goal is anomaly detection. However, as noted before, many anomaly detection methods are not based on forecasting [16, 17].
> > >
> > > Another relevant sub-field is AutoML for anomaly detection (not necessarily for time series). Given a tabular dataset to perform anomaly detection, recent methods do not require labels for this dataset, but they assume availability of a similar dataset with labels (e.g., Kotlar et al. [13], Zhao et al. [14]). In that regard, this methodology is still supervised. We have extended our related work sections (refer Section 6 and Appendix A.1, Meta Learning and Model Selection).
> > >
> > > **References**
> > >
> > > [10] Feurer, Matthias, et al. "Efficient and robust automated machine learning." Advances in neural information processing systems 28 (2015).
> > >
> > > [11] Gijsbers, Pieter, et al. "AMLB: an AutoML Benchmark." arXiv preprint arXiv:2207.12560 (2022).
> > >
> > > [12] Wang, Can, et al. "Towards Time-Series Feature Engineering in Automated Machine Learning for Multi-Step-Ahead Forecasting." Engineering Proceedings 18.1 (2022): 17.
> > >
> > > [13] Kotlar, Miloš, et al. "Novel meta-features for automated machine learning model selection in anomaly detection." IEEE Access 9 (2021): 89675-89687.
> > >
> > > [14] Zhao, Yue, Ryan A. Rossi, and Leman Akoglu. "Automating outlier detection via meta-learning." arXiv preprint arXiv:2009.10606 (2020).

---

> > > > ### Author Response · Authors · 2022-11-17
> > > > **Regarding benchmarks and evaluation strategies (4/4)**
> > > >
> > > > > *Omitted new benchmarks and evaluation strategies*
> > > >
> > > > We thank the reviewer for pointing out relevant work [15] and [16]. We have included these references in the revised manuscript. We do not believe that our conclusions change in light of these studies. **In fact we find [15] and [16] to be supportive of our work.**
> > > >
> > > > For example, Paparrizos et al. [16] conclude “Our findings … demonstrate the difficulty of methods to consistently perform well across such a diverse set of time series and anomaly types”, strengthening the motivation behind our own study.
> > > >
> > > > Next, reference [15] highlights limitations of threshold-based and point-based measures. This is in line with our own understanding (see Section 2.1). Consequently, we used a threshold-independent measure adjusted best-$F_1$, as opposed to pointwise $F_1$. Multiple recent studies [19, 20, 21, 22, 23] have used adjusted best-$F_1$ to evaluate anomaly detection performance. Moreover, we found adjusted best-$F_1$ to be highly correlated with adjusted PR-AUC (spearman correlation = 0.95, kendall-$\tau$ = 0.83 with $p$-value < 1e-3), consistent with prior work [23]. We have added the correlation plot to appendix A.10.
> > > >
> > > > Since the recently proposed Volume under the Surface of the Receiver Operating Curve (ROC) (VUS-ROC) metric [15] was defined and tested on univariate data, we still carry out all experiments using adjusted best $F_1$, but instead evaluate the correlation between VUS-ROC and adjusted best $F_1$ for each time-series in the UCR anomaly archive [18].
> > > >
> > > > We found that out of 250 time-series, **only 23 time-series had neither statistically significant positive kendall-tau nor spearman-r correlation.** All these time-series resulted in low dispersion of the evaluation measures i.e. the difference in performance of most models trained on the time-series, measured in terms of VUS-ROC or adjusted best $F_1$, was less than 10%. Thus lack of correlation in these time series is explained by the fact that models performed on par.
> > > >
> > > > While we appreciate substantial contributions of [16], we believe that the data that we used in our study (SMD [19] and UCR Anomaly archive Wu et al. [18]) is sufficiently diverse to support our conclusions. For example, it covers multiple domains, such as ECG, temperature measurements and IT Ops.
> > > >
> > > > *In light of these changes and new results, we sincerely hope that the reviewer would consider increasing their score.*
> > > >
> > > > **References**
> > > >
> > > > [15] Paparrizos, John, et al. "Volume under the surface: a new accuracy evaluation measure for time-series anomaly detection." Proceedings of the VLDB Endowment 15.11 (2022): 2774-2787.
> > > >
> > > > [16] Paparrizos, John, et al. "TSB-UAD: an end-to-end benchmark suite for univariate time-series anomaly detection." Proceedings of the VLDB Endowment 15.8 (2022): 1697-1711.
> > > >
> > > > [17] Schmidl, Sebastian, Phillip Wenig, and Thorsten Papenbrock. "Anomaly detection in time series: a comprehensive evaluation." Proceedings of the VLDB Endowment 15.9 (2022): 1779-1797.
> > > >
> > > > [18] Wu, Renjie, and Eamonn Keogh. "Current time series anomaly detection benchmarks are flawed and are creating the illusion of progress." IEEE Transactions on Knowledge and Data Engineering (2021).
> > > >
> > > > [19] Su, Ya, et al. "Robust anomaly detection for multivariate time series through stochastic recurrent neural network." Proceedings of the 25th ACM SIGKDD international conference on knowledge discovery & data mining. 2019.
> > > >
> > > > [20] Carmona, Chris U., et al. "Neural contextual anomaly detection for time series." arXiv preprint arXiv:2107.07702 (2021).
> > > >
> > > > [21] Challu, Cristian I., et al. "Deep Generative model with Hierarchical Latent Factors for Time Series Anomaly Detection." International Conference on Artificial Intelligence and Statistics. PMLR, 2022.
> > > >
> > > > [22] Shen, Lifeng, Zhuocong Li, and James Kwok. "Timeseries anomaly detection using temporal hierarchical one-class network." Advances in Neural Information Processing Systems 33 (2020): 13016-13026.
> > > >
> > > > [23] Xu, Haowen, et al. "Unsupervised anomaly detection via variational auto-encoder for seasonal kpis in web applications." Proceedings of the 2018 world wide web conference. 2018.

---

### Official Review · Reviewer_VFqz · 2022-10-24

**Confidence:** 2
**Correctness:** 3
**Technical Novelty And Significance:** 3
**Empirical Novelty And Significance:** 3
**Recommendation:** 6

**Clarity, Quality, Novelty And Reproducibility:**

The idea is interesting and easy to follow, with extensive evaluations on multiple real-world time series anomaly detection datasets.

**Strength And Weaknesses:**

Pros:
1. The proposed model selection criterion is simple and flexible to implement.
2. Extensive experiments using various anomaly detection criteria have been performed.

Cons:
1. In the numerical experiments, the proposed rank aggregation strategy is only applied to five pre-selected anomaly injection metrics. It is suggested to run the rank aggregation strategy on all the studied surrogate metric in order to fully showcase the capability of the proposed anomaly detection model selection criterion.
2. The superiority of the proposed method in comparison to other metrics is not that significant.


**Summary Of The Paper:**

This manuscript tried to answer the question in the community of time series anomaly detection: given an unlabeled dataset and a set of candidate anomaly models, how can we select the most accurate model? Therefore, this paper proposed a robust rank aggregation method with theoretical justifications. This unsupervised model selection method has been verified on multiple real-world datasets.

**Summary Of The Review:**

Overall the idea of using rank aggregation for model selection for time series anomaly detection offers new insights, and the evaluation is solid.

---

> ### Author Response · Authors · 2022-11-17
> **Summary of changes and responses**
>
> > *It is suggested to run the rank aggregation strategy on all the studied surrogate metric*
>
> We thank the reviewer for this suggestion. We have now added results of an experiment where we perform rank aggregation using all the metrics. Detailed results are in Appendix A.12.
>
> > *Superiority of the proposed method in comparison to other metrics is not that significant*
>
> Our main argument is that the proposed minimum influence metric (MIM) is never worse than supervised selection (SS). This is useful, because MIM is fully automated, while SS requires manual labels.

---

### Official Review · Reviewer_Qvoy · 2022-10-26

**Confidence:** 4
**Clarity, Quality, Novelty And Reproducibility:** No issues, everything is well present…
**Correctness:** 3
**Technical Novelty And Significance:** 4
**Empirical Novelty And Significance:** 3
**Recommendation:** 8

**Strength And Weaknesses:**

Strengths:
- Very interesting concept, does not seem well studied prior in the literature
- The proposed Borda rank aggregation has theoretical justifications
- Overall well presented paper

Weaknesses:
- Is minimizing empirical influence, essentially throwing out "bad" surrogate metrics as \sigma_i are the rankings of surrogate metric i ? If so, in the empirical results it would be interesting to see if the surrogate metrics that are "bad" for a dataset, align with that metrics performance on the dataset. I believe the surrogate metrics being aggregated are the same ones used individually?

- Under the assumptions of Theorem 1 the probability that Borda aggregation makes a mistake in ranking is an exponentially decaying in the number of surrogate metrics M. The experimental results never seem to empirically support this though. While it would be potentially computationally expensive to show it asymptotically, it would be good to see a plot of the performance of Borda aggregation increasing as the number of surrogate metrics increase, even if it is just from 2 to 17.

- None of the methods are able to have significant wins against random on all 10 datasets? Additionally Pred. Error MSE has significant losses against Random in 4 datasets; but it is used as one of the surrogate metrics in Borda? For those 4 datasets, wouldn't that metric basically violate the assumptions of Theorem 1 that Borda is aggregating surrogate metrics that are better than random?

**Summary Of The Paper:**

This paper focuses on anomaly detection model selection metrics in the unsupervised setting where a user does not have access to labeled anomalies in order to select the best model. They identify three classes of these surrogate metrics (prediction error, synthetic anomaly injection, and model centrality) and also propose a way to combine rankings from multiple surrogate metrics with some theoretical guarantees. They show experiments comparing surrogate metrics from the three classes with their combined rankings method against an oracle, random baseline, and partially labeled baseline.

**Summary Of The Review:**

This paper is presented well, proposes an interesting novel concept, and has strong justifications. The only weakness is that empirical results seem somewhat mixed in supporting their claims.

---

> ### Author Response · Authors · 2022-11-17
> **Summary of changes and responses**
>
> > *It would be interesting to see if the surrogate metrics that are "bad" for a dataset, align with that metrics performance …*
>
> We thank the reviewer for this suggestion. We have now added results of experiments measuring the rank correlation between empirical influence and quality of a metric on synthetic data (Appendix A.4) and on real-world datasets (Appendix A.11). On synthetic data, we found that empirical influence can almost perfectly identify bad metrics. 7 out of 10 real world datasets demonstrated negative kendall-$\tau$ correlation between the quality of a metric and its empirical influence. Of these 7 datasets, 5 datasets had statistically significant negative correlation. Recall that bad metrics have high positive empirical influence (Section 4.2).
>
> *Edit*:
> Based on your suggestions, we have now also included theoretical results to support our use of empirical influence to identify bad permutations in Appendix A.13.
>
> > *the surrogate metrics being aggregated are the same ones used individually?*
>
> Yes, the surrogate metrics being aggregated are the ones used individually.
>
> > *None of the methods are able to have significant wins against random on all 10 datasets?*
>
> In Table 2 and 3, minimum influence metric (MIM) has 7 out of 10 wins against random. However, our main argument is that the final method that we advocate MIM is never worse than supervised selection (SS). This is useful, because MIM is fully automated, while SS requires manual labels.
>
> > *wouldn't that metric basically violate the assumptions of Theorem 1*
>
> Yes, that might be happening, and that’s why we propose to use MIM, which is based on using empirical influence for pruning the set of metrics.
>
> *We kindly ask the reviewer to consider raising their score in light of these changes and results*

---

> > ### Comment · Reviewer_Qvoy · 2022-12-01
> > **No Address to Bullet 2?**
> >
> > Could you please address bullet 2 of weaknesses to some extent?
> >
> > "Under the assumptions of Theorem 1 the probability that Borda aggregation makes a mistake in ranking is an exponentially decaying in the number of surrogate metrics M. The experimental results never seem to empirically support this though. While it would be potentially computationally expensive to show it asymptotically, it would be good to see a plot of the performance of Borda aggregation increasing as the number of surrogate metrics increase, even if it is just from 2 to 17."
> >
> > The theory behind this method is based on asymptotics that are heavily dependent on a lot of assumptions, waiving away of constants, etc. Some kind of empirical result that reflects that there is at least an improvement when more surrogate metrics are used would at least indicate that the theory is not completely unpractical.

---

> > > ### Author Response · Authors · 2022-12-03
> > > **Thanks for your question, response to bullet 2**
> > >
> > > Thank you for bringing this to our attention. We carried out an experiment based on your suggestion to provide empirical support of our theorem.
> > >
> > > ### Experimental setting
> > > Theorem 1 considers error bounds when ranking two models. Thus, for each time series we identify the best- and worst-performing models and empirically test whether aggregation over more rankings helps reduce this error. We varied the number of surrogate metrics from 2 to 17, and perform borda rank aggregation to find the best model. The performance of the aggregated borda rank is measured as the adjusted best $F_1$ of the top ranked model. For each combination of $m$ metrics, we report the performance averaged over all ${17 \choose m}$ unique combinations of the metrics.
> > >
> > > ### Results
> > > First, we present the performance of the best model as identified by borda aggregation of a combination of metrics, averaged across all the time-series. Overall, **we see a clear trend where the performance of the selected best model increases with the number of surrogate metrics (rankings)**.
> > >
> > > ```python
> > > Avg. perf. of all pairs of 2 metrics: 0.588
> > > Avg. perf. of all pairs of 5 metrics: 0.795
> > > Avg. perf. of all pairs of 8 metrics:  0.784
> > > Avg. perf. of all pairs of 11 metrics: 0.829
> > > Avg. perf. of all pairs of 14 metrics: 0.821
> > > Avg. perf. of all pairs of 17 metrics: 0.838
> > > ```
> > > In only 27 out of 181 cases, we did not observe a monotonic improvement as the number of metrics increased. However, we notice that in all such cases, the performance of all anomaly detection models was low and similar to each other. Under such circumstances, not only do surrogate metrics tend to be noisy, violating the assumptions of the theorem, but also, model selection does not matter much.
> > >
> > > Below, as an example, we also report the results of two individual time-series where we can see our theorem at play, and another time-series where our theorem does not hold:
> > >
> > > ```python
> > > Entity: machine-3-6 (Server Machine Dataset)
> > > Avg. perf. of all pairs of 2 metrics: 0.507
> > > Avg. perf. of all pairs of 5 metrics: 0.701
> > > Avg. perf. of all pairs of 8 metrics: 0.594
> > > Avg. perf. of all pairs of 11 metrics: 0.742
> > > Avg. perf. of all pairs of 14 metrics: 0.629
> > > Avg. perf. of all pairs of 17 metrics: 0.984
> > > ```
> > > ```python
> > > Entity: InternalBleeding17 (UCR anomaly archive)
> > > Avg. perf. of all pairs of 2 metrics: 0.566
> > > Avg. perf. of all pairs of 5 metrics: 0.742
> > > Avg. perf. of all pairs of 8 metrics: 0.645
> > > Avg. perf. of all pairs of 11 metrics: 0.780
> > > Avg. perf. of all pairs of 14 metrics: 0.677
> > > Avg. perf. of all pairs of 17 metrics: 0.999
> > > ```
> > > ```python
> > > Entity: DISTORTEDInternalBleeding17 (UCR anomaly archive)
> > > Avg. perf. of all pairs of 2 metrics: 0.324
> > > Avg. perf. of all pairs of 5 metrics: 0.396
> > > Avg. perf. of all pairs of 8 metrics: 0.272
> > > Avg. perf. of all pairs of 11 metrics: 0.268
> > > Avg. perf. of all pairs of 14 metrics: 0.240
> > > Avg. perf. of all pairs of 17 metrics: 0.240
> > > ```
> > >
> > > We would add these results to the camera ready version of the paper.
> > >
> > > Please let us know if you have more questions.

---

> > > > ### Comment · Reviewer_Qvoy · 2022-12-12
> > > > **Thank you for addressing Bullet 2**
> > > >
> > > > I have updated my score. The new experiments look convincing.

---

> > > > > ### Author Response · Authors · 2022-12-12
> > > > > **Thanks for your time and feedback**
> > > > >
> > > > > Dear Reviewer Qvoy,
> > > > > Thank you so much for your time and feedback. We firmly believe that it helped us improve our work.
> > > > >
> > > > > Best,
> > > > > Authors

---

### Author Response · Authors · 2022-11-17
**Opening Comment and Summary of Changes**

We thank all reviewers for their valuable comments. We are delighted that the reviewers found our work “*quite a fundamental research problem*” (kY2N), and described our approach as a “*very interesting concept*” (Qvoy), "*simple and flexible to implement*" (VFqz), “*the idea [is] interesting*” (VFqz). Some reviewers commented that “*experiments based on many real datasets of various domains*” (kY2N), and that the paper was “*well-written and easy to follow*” (E7ki). We have split our responses into separate messages for reading convenience and to bypass message length limits. We have highlighted new changes to the paper in response to the reviews in blue.

We have made several changes to the paper and carried our 4 experiments based on reviewer suggestions:
1. **Related work**: Based on **reviewer E7Ki’s** suggestions, we have added longer discussions in the related work section, comparing our work with AutoML, papers using forecasting error for forecasting model selection and semi-supervised anomaly detection model selection (see Appendix A.1). We identified several limitations of related work which we summarize towards the end of the related work section.
2. **Rank aggregation using all the metrics**: Here we carry out experiments using all the metrics instead of the top-5 best performing metrics, suggested by **reviewer VFqz** (see Appendix A.12).
3. **Does empirical influence identify bad ranks?**: We measured the correlation between empirical influence and quality of a metric on synthetic data (Appendix A.4) and on real-world datasets (see Appendix A.11), based on suggestions by **reviewer Qvoy**.
4. **Correlation between adjusted best $F_1$ and VUS-ROC**: Based on **reviewer E7ki’s** suggestion, we now report the correlation between adjusted best $F_1$, the metric we use in this paper, and VUS-ROC (see Appendix A.9).
5. **Correlation between adjusted best $F_1$ and adjusted PR-AUC**: We also compared the correlation between adjusted best $F_1$ and PR-AUC (see Appendix A.10).
6. **Theoretical justification of Empirical Influence**: Based on **reviewer Qvoy's** concerns, we have now included theoretical justification to support our use of Empirical Influence to identify bad permutations (see Appendix A.13).

We are still carrying out experiments and will continue to update results as we get them. *In light of these changes and new results, we sincerely hope that the reviewers consider increasing their scores.*

---

> ### Author Response · Authors · 2022-11-18
> **More theoretical results**
>
> We would like to thank the reviewers for their time and excellent suggestions to improve our work.
>
> Based on reviewer Qvoy's suggestion, we have now included theoretical results to justify how empirical influence identifies bad permutations in Appendix A.13. We have also edited our previous rebuttal responses and manuscript revision accordingly.

---

### Public Comment · ~Prabhant_Singh2 · 2023-05-05
**Code request**

Hi as the paper is accepted, I wanted to know where can I find the code for this work?

---

> ### Author Response · Authors · 2023-05-05
> **Code is available on Github**
>
> Hi,
> Thanks for your interest! The code is available at `https://github.com/mononitogoswami/tsad-model-selection/`.
> We'll continue to make changes to it to ensure reproducibility of our results.
>
> Thanks,
> Authors

---

### Decision · Program_Chairs · 2023-01-20

**Decision:**

Accept: notable-top-25%

**Justification For Why Not Higher Score:**

- Method is of specific interest to the anomaly detection community and does not introduce any new, broadly applicable techniques (is a combination of existing ideas)

**Justification For Why Not Lower Score:**

- Paper addresses an important problem with a well-motivated and justified method
- Proposed method is general and easy to implement
- Extensive experiments on real-world datasets are provided to support the method

**Metareview: Summary, Strengths And Weaknesses:**

This paper proposes an unsupervised selection method for time-series anomaly detection algorithms. Reviewers thought that the paper addresses an important problem with a well-motivated approach, that the method was easy to implement, and appreciated the extensive experiements performed. However, reviewers had concerns regarding novelty of the work and the superiority of the proposed method. The authors provided an extensive discussion of related work in their response and clarified the advantages of the proposed method (it does not need labels); they also included additional results to strengthen the justification of their method, which led to a score increase. Overall, the AC agrees that the problem addressed is an important one, the proposed method is well-motivated and supported by extensive experimentation, and thinks that the concerns on novelty are adequately addressed in the response. As such, the AC recommends acceptance.

**Note From Pc:**

if the above contains the word "oral" or "spotlight" please see: "oral" presentation means -> notable-top-5% and "spotlight" means -> notable-top-25%. As stated in our emails, we are disassociating presentation type from AC recommendations